# Incentivizing High-quality Participation From Federated Learning Agents

## Abstract

Federated learning (FL) provides a promising paradigm for facilitating collaboration between multiple clients that jointly learn a global model without directly sharing their local data. However, existing research suffers from two caveats: 1) From the perspective of agents, voluntary and unselfish participation is often assumed. But self-interested agents may opt out of the system or provide low-quality contributions without proper incentives; 2) From the mechanism designer's perspective, the aggregated models can be unsatisfactory as the existing game-theoretical federated learning approach for data collection ignores the potential heterogeneous effort caused by contributed data. To alleviate above challenges, we propose an incentive-aware framework for agent participation that considers data heterogeneity to accelerate the convergence process. Specifically, we first introduce the notion of Wasserstein distance to explicitly illustrate the heterogeneous effort and reformulate the existing upper bound of convergence. To induce truthful reporting from agents, we analyze and measure the generalization error gap of any two agents by leveraging the peer prediction mechanism to develop score functions. We further present a two-stage Stackelberg game model that formalizes the process and examines the existence of equilibrium. Extensive experiments on real-world datasets demonstrate the effectiveness of our proposed mechanism.

## 1 Introduction

Real-world datasets are often spread across multiple locations, and privacy concerns can prevent centralizing this data for training. Federated Learning (FL) provides a promising paradigm for facilitating collaboration among multiple agents (or clients) to jointly learn a global model without sharing their local data. In the standard FL framework, each agent trains a model using its own data, and a central server aggregates these local models into a global one. Despite the rapid growth and success of FL in improving speed, efficiency, and accuracy (Li et al., 2019; Yang et al., 2021), many studies assume, often unrealistically, that agents will voluntarily spend their resources on collecting their local data and training models to help the central server to refine the global model. In practice, without proper incentives, self-interested agents may opt out of contributing their resources to the system or provide low-quality models (Liu & Wei, 2020; Blum et al., 2021; Wei et al., 2021; Karimireddy et al., 2022).

Recent literature has observed efforts to incentivize FL agents to contribute model training using sufficient local data (Donahue & Kleinberg, 2021b;a; Hasan, 2021; Cui et al., 2022; Cho et al., 2022). One **common limitation** of these studies is that they measure each agent's contribution simply by the sample size used for training the uploaded model and incentivize agents to use more data samples. However, relying solely on the sample size to measure contribution may not accurately capture the quality of data samples. This is because aggregating local models trained from large data does not always lead to fast convergence or highly accurate global models if those data are biased (McMahan et al., 2017; Zhao et al., 2018). In practice, local data on different agents are heterogeneous, also referred to as non-independently and identically distributed (non-iid). Hence, one appropriate incentivized method should consider not just the sample size but also the potential data heterogeneity. Such a solution is necessary for developing incentive mechanisms in practical FL systems but is currently missing from existing work (Blum et al., 2021; Liu & Wei, 2020; Karimireddy et al., 2022; Zhou et al., 2021; Donahue & Kleinberg, 2021a;b; Yu et al., 2020; Cho et al., 2022). In this

work, we seek to answer the following pertinent question: *How do we incentivize agents (local clients) to provide models trained on high-quality data that enable fast convergence on a highly accurate global model?*

To answer this, we propose an incentive-aware FL framework that considers data heterogeneity to accelerate the convergence process. Our main contributions are as follows:

- We propose to use Wasserstein distance to quantify the non-iid degree of the agent's local data. We prove that Wasserstein distance is an important factor in the convergence bound of FL with heterogeneous data (Theorem 2).
- We present a two-stage Stackelberg game model for the incentive-aware framework, and prove the existence of equilibrium (Theorem 5). In particular, to induce truthful reporting, we design scoring (reward) functions to reward agents using a well-tailored metric to quantify the generalization loss gap of any two agents (Theorem 3).
- Experiments on real-world datasets demonstrate the effectiveness of the proposed mechanism in terms of its ability to incentivize improvement. We further illustrate the existing equilibrium that no one can unilaterally reduce his invested effort level to obtain a higher utility.

## 2 Related Work

Significant efforts have been made to tackle the incentive issues in FL. However, those works heavily overlook the agents' strategic behaviors or untruthful reporting problems induced by insufficient incentives (Pang et al., 2022b; Zhou et al., 2021; Yu et al., 2020). We summarize existing studies on this game-theoretic research.

**Coalition game** Recent efforts (Donahue & Kleinberg, 2021a;b; Cho et al., 2022; Hasan, 2021; Cui et al., 2022) have sought to conceptualize federated learning through the lens of coalition games involving self-interested agents, i.e., exploring how agents can optimally satisfy their incentives defined differently from our goal. Rather than focusing on training a single global model, they consider the scenario where each client seeks to identify the most advantageous coalition of agents to federate, and aim to minimize its error.

**Contribution evaluation** Contribution quantification has been extensively explored, both within specific areas of distributed learning (Pang et al., 2022a; 2023; Hua et al., 2024; Han et al., 2021; Zhou et al., 2022) and from a data-centric perspective (Liu et al., 2024; Pang et al., 2024b; 2025; Zhang et al., 2025). In federated learning (FL), a growing body of work focuses on incentivizing agents to contribute resources and ensuring long-term participation by offering monetary compensation based on their contributions, often determined through game-theoretic approaches (Han et al., 2022; Kang et al., 2019; Zhang et al., 2021; Khan et al., 2020). Certain prior studies have attempted to quantify the contributions of agents using metrics like accuracy or loss per round. However, these works fail to analyze model performance theoretically, with some neglecting the effects of data heterogeneity on models (Kong et al., 2022) and others overlooking the impact of samples on data distributions (Xu et al., 2021; Wang et al., 2020). Our work provides a theoretical link between data heterogeneity and models, offering insights into data heterogeneity and helping us handle data diversity.

**Data sharing/collection** Having distinct objectives, recent work strives to directly maximize the number of satisfied agents utilizing a universal model (Cho et al., 2023a;b). In contrast, another work explores an orthogonal setting, where each client seeks to satisfy its constraint of minimal expected error while limiting the number of samples contributed to FL (Blum et al., 2021; Karimireddy et al., 2022). While these works establish insights for mean estimation and linear regression problems, their applicability to complex ML models is limited. Also, these works neglect the discrepancy of data distributions among different agents. We address these limitations by evaluating the non-iid degree of the data contributed by agents and creating score functions to quantify agents' contributions.

## 3 Problem Formulation

### 3.1 Preliminaries

**The distributed optimization model** Consider a distributed optimization model with $F_k : \mathbb{R}^d \to \mathbb{R}$, such that $\min_{\mathbf{w}} \quad F(\mathbf{w}) \triangleq \sum_{k=1}^{N} p_k F_k(\mathbf{w})$, where $\mathbf{w}$ is the model parameters to be optimized, $N$ denotes the number of agents, $p_k$ indicates the weight of the $k$-th agent such that $p_k \geq 0$ and $\sum_{k=1}^{N} p_k = 1$. Each agent participates in training with the local objective $F_k(\boldsymbol{w}) \triangleq \mathbb{E}_{\zeta^k \in \mathcal{D}_k}[\ell(\mathbf{w}, \zeta^k)]$, where $\ell(\cdot, \cdot)$ represents the loss function, and datapoint $\zeta^k = (\boldsymbol{x}, y)$ uniformly sampled from agent $k$'s local dataset $\mathcal{D}_k$.

**Assumptions** Here, we present some generalized basic assumptions in convergence analysis. Note that the first two assumptions are standard in convex/non-convex optimization (Li et al., 2019; Yang et al., 2022; Wang et al., 2019; Koloskova et al., 2020; Pang et al., 2024a). The assumptions of bounded gradient and local variance are both standard (Yang et al., 2021; 2022; Stich, 2018).

**Assumption 1.** *($\mu$-Strongly Convex). There exists a constant $\mu > 0$, for any $\mathbf{v}, \mathbf{w} \in \mathbb{R}^d$, $F_k(\mathbf{v}) \geq F_k(\mathbf{w}) + \langle \nabla F_k(\mathbf{w}), \mathbf{v} - \mathbf{w} \rangle + \frac{\mu}{2} \|\mathbf{v} - \mathbf{w}\|_2^2, \forall k \in [N]$.*

**Assumption 2.** *(L-Lipschitz Continuous). There exists a constant $L > 0$, for any $\mathbf{v}, \mathbf{w} \in \mathbb{R}^d$, $F_k(\mathbf{v}) \leq F_k(\mathbf{w}) + \langle \nabla F_k(\mathbf{w}), \mathbf{v} - \mathbf{w} \rangle + \frac{L}{2} \|\mathbf{v} - \mathbf{w}\|_2^2, \forall k \in [N]$.*

**Assumption 3.** *(Bounded Local Variance). The variance of stochastic gradients for each agent is bounded: $\mathbb{E}[\|\nabla F_k(\mathbf{w}; \zeta^k) - \nabla F_k(\mathbf{w})\|^2] \leq \sigma_k^2, \forall k \in [N]$.*

**Assumption 4.** *(Bounded Gradient on Random Sample). The stochastic gradients on any sample are uniformly bounded, i.e., $\mathbb{E}[\|\nabla F_k(\mathbf{w}_t^k; \zeta^k)\|^2] \leq G^2, \forall k \in [N]$, and epoch $t \in [0, \cdots, T-1]$.*

**Quantifying the non-iid degree (heterogeneous effort)** Empirical evidence (McMahan et al., 2017; Zhao et al., 2018) revealed that the intrinsic statistical challenge of FL, *i.e.*, data heterogeneity, will result in reduced accuracy and slow convergence of the FL global model. However, many prior works assume to bound the gradient dissimilarity with constants, which did not explicitly illustrate this heterogeneous effort (Wang et al., 2019; Koloskova et al., 2020). To quantify the degree of non-iid, we consider discretizing the entire data distribution space into a series of component distributions. In this work, we use the concept of *Wasserstein distance* to capture the inherent heterogeneous gap, aligning with the data collection target. Specifically, the Wasserstein distance $\delta$ is defined as the one-dimensional probability distance between the discrete data distribution $p^{(k)}$ of agent $k$ and the centralized reference (iid) data distribution $p^{(c)}$:

$$\delta_k = \frac{1}{2} \sum_{i=1}^{I} |p^{(k)}(y = i) - p^{(c)}(y = i)|, \tag{1}$$

where $\delta_k \in [0, 1]$ also natural describes the "cost" of transforming agent $k$'s data distribution into the reference distribution, $I$ denotes the number of component distributions, and $p^{(k)}(y = i)$ is the statistical proportion of $i$-th component distribution in agent $k$'s dataset. For better understanding, $I$ can be viewed as the number of labels, with the data distribution being discretized according to these labels. Let $\boldsymbol{\delta} = \{\delta_1, ..., \delta_N\}$ be the set of non-iid degrees for all agents.

**Peer prediction mechanism** Peer prediction is a widely used solution concept for information elicitation, which primarily focuses on developing scoring rules to incentivize or elicit self-interested agents' responses about an event (Miller et al., 2005; Shnayder et al., 2016; Liu & Wei, 2020). Specifically, it works by scoring each agent based on the responses from other agents, leveraging the stochastic correlation between different agents' information to ensure that truth-telling is a strict Bayesian Nash Equilibrium. For instance, in crowdsourcing tasks where human-generated responses cannot be directly verified by ground truth due to its unavailability or the high cost of obtaining it, the goal is to encourage truthful reporting. In the context of FL, each agent's model update is treated as a response, but due to data heterogeneity or malicious behavior, these updates may deviate from the ideal model trained on IID data. Given these uncontrollable behaviors, we borrow the idea from the peer prediction mechanism to develop the scoring functions, as detailed next.

### 3.2 Game-theoretic Formulation

We start with a standard FL scenario which consists of the learner and a set of agents. For the learner, it is natural to aim for faster model convergence by handling data heterogeneity. Therefore, the learner is willing to design well-tailored payment functions to evaluate agents' contributions and provide appropriate rewards. Then, agents strive to attain maximum utility in this scenario. We describe the interactions as a game with two players:

- *Learner*: seeking to train a classifier that endeavors to encourage agents to decrease the non-iid degree. Thereby, the learner attains faster convergence at a reduced cost. To accomplish this, the learner incentivizes agents with the payment (reward) function for their efforts.
- *Agent*: investing its effort $e_k$ in gathering more data samples before training to maximize its utility. Given certain efforts $e_k$, the original data distribution $p^{(k)}$ can become more homogeneous. In practice, each agent incurs a cost while investing its effort.

Here, we formally formulate the connection between non-iid degree $\delta_k$ with the agent's effort $e_k \in [0, 1]$ as a *non-increasing* function $\delta_k(e_k) : [0, 1] \to [0, 1]$. For example, $e_k = 0$ indicates that agent $k$ will keep current data; $e_k = 1$ means agent $k$ makes the maximum effort to ensure data homogeneity. In the game formulated above, the learner's payment functions and objectives are available to all players and are considered common knowledge. Specifically, the statistical information of the reference IID data distribution (e.g., label proportions $p^{(k)}(y = i), \forall i \in [I]$) is accessible because the broadcasted IID criteria serve as a target for agents to make efforts towards. Recall that the final payment function will ensure that agents who make efforts receive non-negative utility and achieve maximum utility through truthful reporting. Following this, we present clear definitions of the utility/payoff functions and the optimization problems.

**Definition 1.** *(Agent's Utility). For agent $k$, its utility can be denoted by $u_k(e_k) \triangleq \mathrm{Payment}_k(e_k) - \mathrm{Cost}_k(e_k), \forall k \in [N]$, where $\mathrm{Payment}_k(e_k)$ is the payment function, and $\mathrm{Cost}_k(e_k)$ is agent $k$'s cost function. Intuitively, the payment and cost functions increase with effort coefficient $e_k$.*

**Definition 2.** *(The learner's Payoff). For a learner, the payoff of developing an FL model is $\mathrm{Payoff} \triangleq \mathrm{Reward}(\boldsymbol{e}) - \sum_{k=1}^{N} \mathrm{Payment}_k(e_k)$, where $\mathrm{Reward}(\boldsymbol{e})$ is the reward given the global model's performance under a set of agents' effort levels $\boldsymbol{e} = \{e_1, e_2, \cdots, e_N\}$. Intuitively, the faster the global model converges, the more reward.*

**Problem 1** (*Maximization of agent's utility*). For any agent $k$, its goal is to maximize utility through $\max_{e_k \in [0,1]} \quad u_k(e_k) = \mathrm{Payment}_k(e_k) - \mathrm{Cost}_k(e_k)$.

**Problem 2** (*Maximization of learner's payoff*). The learner's goal is to maximize its payoff conditioned on Incentive Compatibility (IC), *i.e.*, satisfying the utility of agents are non-negative. The optimization problem is

$$\max_{\boldsymbol{e}} \quad \mathrm{Payoff}, \quad \text{s.t. } u_k(e_k) \geq 0, e_k \in [0, 1], \forall k \in [N].$$

Note that agents invest their efforts once, with their utilities and the learner's payoff being determined before training. Although the utility/payoff functions are conceptually straightforward, the tough task is to establish connections between agents and the learner via the non-iid metric. In the proceeding section, we shall delve into the theoretical aspects to uncover potential connections.

## 4  Preparation: Convergence Analysis of FL

In this section, we rethink and reformulate the existing upper bound of convergence via the Wasserstein distance metric $\delta_k$ (Example 1), which mathematically connects agents and the learner, implicitly allowing one to manipulate its non-iid degree to speed up the global model's convergence. Then, we analyze and measure the generalization loss gap between any two agents.

### 4.1 Rethinking Convergence Bound

Recent works extensively explore the convergence bounds of FL under various settings, such as partial/full agent participation with a specific non-iid metric (Li et al., 2019), bounded gradient/Hessian dissimilarity (Karimireddy et al., 2020), B-local dissimilarity (Li et al., 2020), uniformly bounded variance (Yu et al., 2019), asynchronous communications & different local steps per agent (Yang et al., 2022). However, the generic form remains similar. The fundamental idea is to leverage the following Lemma 1 to derive the convergence results.

**Lemma 1.** *(Lemma 3.1 in (Stich, 2018)). Let $\left\{\mathbf{w}_t^k\right\}_{t \geq 0}$ be defined as the parallel sequences of epochs obtained by performing SGD, and $\overline{\mathbf{w}}_t = \frac{1}{N} \sum_{k=1}^{N} \mathbf{w}_t^k$. Let $\overline{\mathbf{g}}_t = \sum_{k=1}^{N} p_k \nabla F_k(\mathbf{w}_t^k)$ represent the weighted sum of gradients of loss functions $F_k$ for agent $k$. $\mathbf{g}_t = \sum_{k=1}^{N} p_k \nabla F_k(\mathbf{w}_t^k, \xi_t^k)$. Let $F(\cdot)$ be L-smooth and $\mu$-strongly convex and the learning rate $\eta_t \leq \frac{1}{4L}$. Then,*

$$\mathbb{E}\left\|\overline{\mathbf{w}}_{t+1} - \mathbf{w}^\star\right\|^2 \leq (1 - \mu\eta_t)\mathbb{E}\left\|\overline{\mathbf{w}}_t - \mathbf{w}^\star\right\|^2 + \eta_t^2 \mathbb{E}\left\|\mathbf{g}_t - \overline{\mathbf{g}}_t\right\|^2 - \frac{1}{2}\eta_t \mathbb{E}\left(F\left(\overline{\mathbf{w}}_t\right) - F(\mathbf{w}^*)\right)$$
$$+ 2\eta_t \underbrace{\mathbb{E}[\sum\nolimits_{k=1}^{N} p_k || \overline{\mathbf{w}}_t - \mathbf{w}_t^k ||^2]}_{Divergence\ term}.$$

The *divergence term* implicitly encodes the impact of data heterogeneity due to the observation that the discrepancy of models incurred by data heterogeneity will be shown up as the divergence of model parameters (Zhao et al., 2018). Therefore, to build a connection between convergence and the Wasserstein distance metric, the first step is to analyze the divergence term. Here, we present the main result of our paper, given in Theorem 2.

**Theorem 2.** *(Bounded the divergence of $\{\mathbf{w}_t^k\}$). Let Assumption 4 hold and G be defined therein, given the synchronization interval E (local epochs), the learning rate $\eta_t$. Suppose $\nabla_{\mathbf{w}}\mathbb{E}_{\boldsymbol{x}|y=i}[\ell_i(\boldsymbol{x}, \mathbf{w})]$ is L-Lipschitz, it follows that*

$$\mathbb{E}[\sum_{k=1}^{N} p_k || \overline{\mathbf{w}}_t - \mathbf{w}_t^k ||^2] \leq 16(E-1)G^2\eta_t^2(1 + 2\eta_t L)^{2(E-1)}\Psi$$

*where $\Psi = \sum_{k=1}^{N} p_k \underbrace{\left(\sum_{i=1}^{I} |p^{(k)}(y=i) - p^{(c)}(y=i)|\right)^2}_{4\delta_k^2}.$*

Using Theorem 2, existing convergence results could be simply rewritten by substituting the upper bound of the divergence term with its new form. Here, we introduce an illustrative example (Example 1) to highlight the convergence results, which help develop our utility/payoff function in the next section. We also use this running example as our setting throughout the paper.

**Example 1.** *(Theorem 1 in (Li et al., 2019)). Given Assumptions 1 to 3, suppose $\sigma_k^2$, G, $\mu$, L are defined therein, the synchronization interval E (local epochs), $\kappa = \frac{L}{\mu}$, $\gamma = \max\{8\kappa, E\}$ and the learning rate $\eta_t = \frac{2}{\mu(\gamma+t)}$. FedAvg with full device participation follows*

$$\mathbb{E}[F(\mathbf{w}_T)] - F^* \leq \frac{\kappa}{\gamma + T - 1}\left(\frac{2B}{\mu} + \frac{\mu\gamma}{2}\mathbb{E}||\mathbf{w}_1 - \mathbf{w}^*||^2\right).$$

*where $B = \underbrace{16(E-1)G^2\eta_t^2(1 + 2\eta_t L)^{2(E-1)}\sum_{k=1}^{N} p_k\delta_k^2}_{Upper\ bound\ shown\ in\ Theorem\ 2} + \sum_{k=1}^{N} p_k^2\sigma_k^2 + 6L\Gamma.$*

In Example 1, the upper bound of convergence (aka, model performance) is affected by the non-iid metric $\delta_k^2$, which means one builds a mathematical link between agents and the learner, which implicitly allows one to manipulate its non-iid degree to speed up the global model's convergence. The result highlighted in

Example 1 gives us the certainty that the non-iid metric can be effectively utilized to develop the payoff function. Note that we focus solely on the divergence term for analysis, as exploring the other terms in the convergence results is beyond our scope. For interested readers, please refer to (Li et al., 2019) for more details.

**Remark 1.** *(Tightness) Leveraging this generic form, our results maintain broad applicability without compromising the tightness of existing bounds. Incorporating our results (Theorem 2) into existing results (e.g., Example 1) would not diminish the tightness of original bounds, ensuring they remain as tight as the original bounds. For instance, compared to Example 1's upper bound of the divergence term $(4\eta_t^2(E-1)^2G^2)$, our bound shown in Theorem 2 $(64\delta_k^2(1+2\eta_tL)^{2(E-1)}\eta_t^2(E-1)G^2)$ aligns consistently.*

**Connections with the assumption that bounds global gradient variance** Note that there exists a common assumption in (Yang et al., 2021; 2022; Wang et al., 2019; Stich, 2018) about bounding gradient dissimilarity or variance using constants. In AppendixA.7, to illustrate the broad applicability of the proposed non-iid degree metric, we further discuss how to rewrite this type of assumption by leveraging $\delta_k$, that is, re-measuring the gradient discrepancy among agents, *i.e.*, $\|\nabla F_k(\mathbf{w}) - \nabla F(\mathbf{w})\|^2, \forall k \in [N]$.

### 4.2 Generalization Loss Gap Between Peers

Note that in FL, the learner only receives the responses provided by agents (model $\mathbf{w}$) without any further information. To promote truthful reporting, we consider constructing utility functions via peer prediction mechanism (Liu & Wei, 2020; Shnayder et al., 2016). Suppose that the learner owns an auxiliary/validation dataset $\mathcal{D}_c = \{(x_n, y_n)\}_{n=1}^N$ following a reference data distribution $p^{(c)}$. To start with, we consider *one-shot* setting that agent $k$, $k'$ submit their models only in a single communication round $E$.

**Theorem 3.** *Suppose that the expected loss function $F_c(\cdot)$ follows Assumption 2, then the upper bound of the generalization loss gap between agent $k$ and $k'$ is*

$$F_c(\mathbf{w}^k) - F_c(\mathbf{w}^{k'}) \leq \Phi\delta_k^2 + \Phi\delta_{k'}^2 + \Upsilon \tag{2}$$

*where $\Phi = 16L^2G^2 \sum_{t=0}^{E-1}(\eta_t^2(1+2\eta_t^2L^2))^t$, $\Upsilon = \prod_{t=0}^{E-1}(1-2\eta_tL)^t \frac{2G^2L}{\mu^2} + \frac{LG^2}{2}\sum_{t=0}^{E-1}(1-2\eta_tL)^t\eta_t^2$, and $F_c(\mathbf{w}) \triangleq \mathbb{E}_{z\in\mathcal{D}_c}[\ell(\mathbf{w},z)]$ denotes the generalization loss induced when the model $\mathbf{w}$ is evaluated on dataset $\mathcal{D}_c$.*

Theorem 3 postulates that the upper bound of the generalization loss gap is intrinsically associated with the degrees of non-iid $\delta_k^2$ and $\delta_{k'}^2$. Inspired by this, we propose to utilize the generalization loss gap between two agents for the incentive design of the scoring function.

## 5 Equilibrium Characterization

The preceding findings (Theorem 2 & Theorem 3) provide insights for the incentive design of the mechanism, which endeavors to minimize the Wasserstein distance, thereby expediting the convergence process artificially. These observations motivate us to formulate our utility/payoff functions, and present a two-stage Stackelberg game model that formalizes the incentive process and examines the existence of equilibrium in federated learning.

### 5.1 Payment Design

**Agents' utilities** Recall that strategic behaviors across agents potentially exist such as free-riding if one can maximize their utilities (Karimireddy et al., 2022). To address this, inspired by the peer prediction (Shnayder et al., 2016; Liu & Wei, 2020), we introduce randomness and evaluate the contribution of agents by quantifying the performance disparity (i.e., generalization loss gap) between agent $k$ and a *randomly* selected peer agent $k'$. The use of randomness in peer prediction can effectively circumvent the coordinated strategic behaviors of agents and elicit truthful information from individuals without ground-truth verification, by leveraging peer responses assigned for the same task. Then, based on the generalization loss gap (Theorem

3), we develop a scoring rule (*a.k.a.*, payment functions) to measure agents' contributions:

$$\text{Payment}_k(e_k, e_k') \triangleq f\left(\frac{Q}{\Phi\delta_k^2(e_k) + \Phi\delta_{k'}^2(e_{k'}) + \Upsilon}\right) \propto \frac{1}{\text{Upper\_Bound}(F_c(\mathbf{w}^k) - F_c(\mathbf{w}^{k'}))}$$

where $Q$ denotes the coefficient determined by the learner, *a.k.a.*, initial payment. Basically, the payment function $f : \mathbb{R}^+ \to \mathbb{R}^+$ can be a monotonically non-decreasing function and inversely proportional to Upper\_Bound$(F_c(\mathbf{w}^k) - F_c(\mathbf{w}^{k'}))$, which indicates that the smaller generalization loss gap between one and randomly picked peer agent, the more reward. The cost for producing data samples to narrow down the non-iid degree of their local data can be defined as:

$$\text{Cost}_k(e_k) \triangleq c \cdot d(|\delta_k(0) - \delta_k(e_k)|), \quad \forall k \in [N].$$

where $c$ means the identical marginal cost for producing data samples, $\delta_k(0)$ and $\delta_k(e_k)$ denotes the Wasserstein distance under effort coefficient $e_k$ is 0 and $e_k$, respectively. $d(\cdot)$ function scales the distance to reflect the collected sample size, which potentially returns a larger value when $\delta_k(e_k)$ is close to $\delta_k(0)$. Note that $\delta_k(0)$ signifies the initial constant value that reflects the original heterogeneous level. For example, in the data-sharing scenario (Blum et al., 2021; Karimireddy et al., 2022), one can think that agent $k$ does not own any data samples yet, and therein $\delta_k(0) = 1/2$.

**Exploring why only adding examples lessens non-iid degree** From the perspective of agents, they can be motivated to strategically provide data samples for minority classes rather than for majority classes in their local datasets. In practice, the influence of training data size is hinted in the convergence bound (Example 1) via a common assumption (Assumption 3). With fewer samples, expected local variance bounds increase, harming model performance. Thus, the learner can evaluate the value of $\delta_k$ and adjust the coefficient $Q$ in the payment function, preventing agents from reducing majority class data samples to attain IID.

**The learner's payoff** Given the convergence result shown in Theorem 2 and Example 1, we can exactly define the payoff function of the learner as follows.

$$\text{Payoff} \triangleq g\left(\frac{1}{\text{Bound}(\boldsymbol{e})}\right) - T\sum_{k=1}^{N} f(e_k, e_{k'}).$$

where Bound$(\boldsymbol{e})$ denotes the model's performance upper bound under agents' efforts $\boldsymbol{e}$ and function $g : \mathbb{R}^+ \to \mathbb{R}^+$ can be a monotonically non-decreasing function and inversely proportional to Bound$(\boldsymbol{e})$ representing that the tighter upper bound, the more reward. $T$ indicates the number of communication rounds if we recalculate and reward agents per round. Notice that Bound$(\boldsymbol{e})$ can be simplified as $\Omega + \gamma\sum_{k=1}^{N} p_k\delta_k^2(e_k)$, and $\Omega$ can be regarded as a constant from Example 1.

## 5.2 Two-stage Game

The interactions between agents and the learner can be formalized as a two-stage Stackelberg game. In Stage I, the learner strategically determines the reward policy (payment coefficient $Q$) to maximize its payoff. In Stage II, each agent $k$ selects its effort level $e_k$ to maximize its own utility. After completing the interactions, the learner would accordingly select agents for training. To obtain an equilibrium, we employ the method of backward induction.

1ex **Agents' best responses** For any agent $k$, he will make efforts with effort level $e_k \neq 0$ if $u(e_k, e_{k'}) \geq u(0, e_{k'})$. This motivates the optimal effort level for the agent. Before diving into finding the existence of equilibrium, we make a mild assumption on payment function $f(\cdot)$:

**Assumption 5.** $f(\cdot)$ *is non-decreasing, differentiable, and strictly concave on* $\boldsymbol{e} \in [0, 1]$.

This assumption states that the speed of obtaining a higher payment is decreasing. Note that this assumption is natural, reasonable, and aligns with the Ninety-ninety rule (Bentley, 1984). The rule states that the first 90% of the work in a project takes 10% of the time, while the remaining 10% of the work takes the other

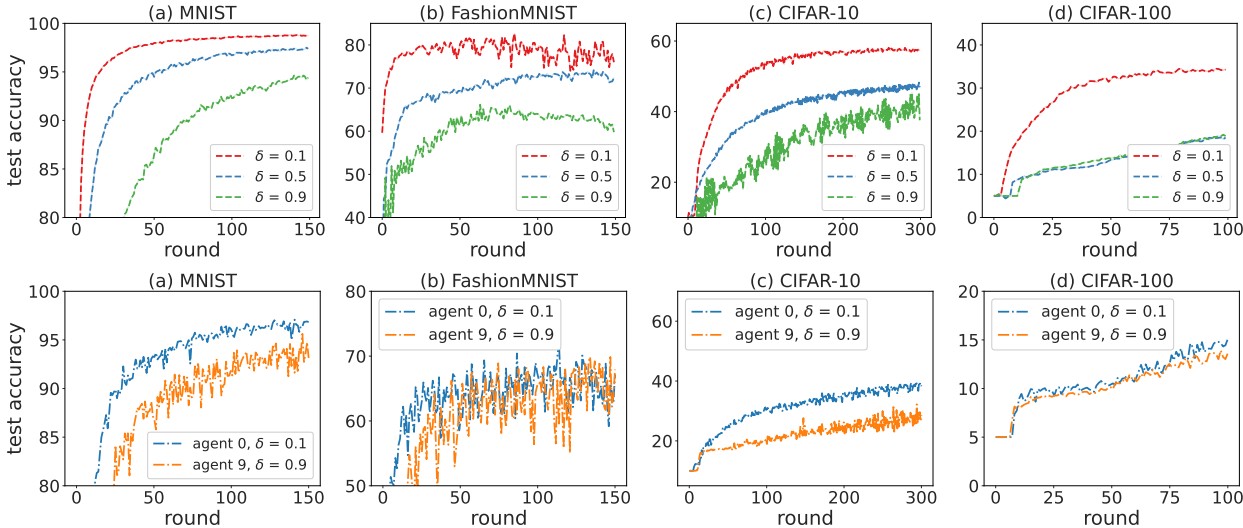

Figure 1: **Upper**: FL training process under different non-iid degrees; **Lower**: Performance comparison with peers.

90% of the time. Several commonly used functions, such as the logarithmic function, conform to this mild assumption. Then, in AppendixA.8, we demonstrate how the payment function with a logarithmic form holds Assumption 5.

Let us consider a hypothetical scenario where an image classification problem entails the classification of ten classes. In such a case, the agent $k$'s dataset is only sampled from one specific class. It is evident that the Wasserstein distance $\delta_k$ decreases rapidly at the initial stages if agent $k$ invests the same effort. Given the inherent nature of this phenomenon, we assume that it is a well-known characteristic of the payment function for both agents and the learner.

**Theorem 4.** *(Optimal effort level). Consider agent $k$ with its marginal cost and the payment function inversely proportional to the generalization loss gap with any randomly selected peer $k'$. Then, agent $k$'s optimal effort level $e_k^*$ is[1]:*

$$e_k^* = \begin{cases} 0, & \text{if } \max_{e_k \in [0,1]} u_k(e_k, e_{k'}) \leq 0; \\ \hat{e}_k & \text{where } \partial u_k(\hat{e}_k, e_{k'})/\partial e_k = 0, \text{otherwise.} \end{cases}$$

The subsequent section will demonstrate that the establishment of an equilibrium in this two-stage game is contingent upon the specific nature of the problem's configuration. Specifically, we shall prove that an equilibrium solution prevails when the impact of unilateral deviations in an agent's effort level remains limited to their own utility. On the other hand, an equilibrium solution may not exist if infinitesimally small changes to an agent's effort level have an outsized effort on other agents' utilities. The characteristics are formalized into the following definition.

**Definition 3.** *(Well-behaved utility functions (Blum et al., 2021)). A set of utility functions $\{u_k : \boldsymbol{e} \to \mathbb{R}^+ | k \in [N]\}$ is considered to be well-behaved over a convex and compact product set $\prod_{k \in [N]}[0,1] \subseteq \mathbb{R}^N$, if and for each agent $k \in [N]$, there are some constants $d_k^1 \geq 0$ and $d_k^2 \geq 0$ such that for any $\boldsymbol{e} \in \prod_{k \in [N]}[0,1]$, for all $k' \in [N]$, and $k' \neq k$, $0 \leq \partial u_k(\boldsymbol{e})/\partial e_{k'} \leq d_{k'}^1$, and $\partial u_k(\boldsymbol{e})/\partial e_k \geq d_k^2$.*

**Theorem 5.** *(Existence of pure Nash equilibrium). Denote by $\boldsymbol{e}^*$ the optimal effort level, if the utility functions $u_k(\cdot)s$ are well-behaved over the set $\prod_{k \in [N]}[0,1]$, then there exists a pure Nash equilibrium in effort level $\boldsymbol{e}^*$ which for any agent $k$ satisfies,*

$$u_k(e_k^*, \boldsymbol{e}_{-k}^*) \geq u_k(e_k, \boldsymbol{e}_{-k}^*), \quad \forall e_k \in [0,1], \forall k \in [N].$$

---

[1]To explicitly highlight the impact of random peer agents, when there is no confusion, we use $u_k(e_k, e_{k'})$ to substitute $u_k(e_k)$.

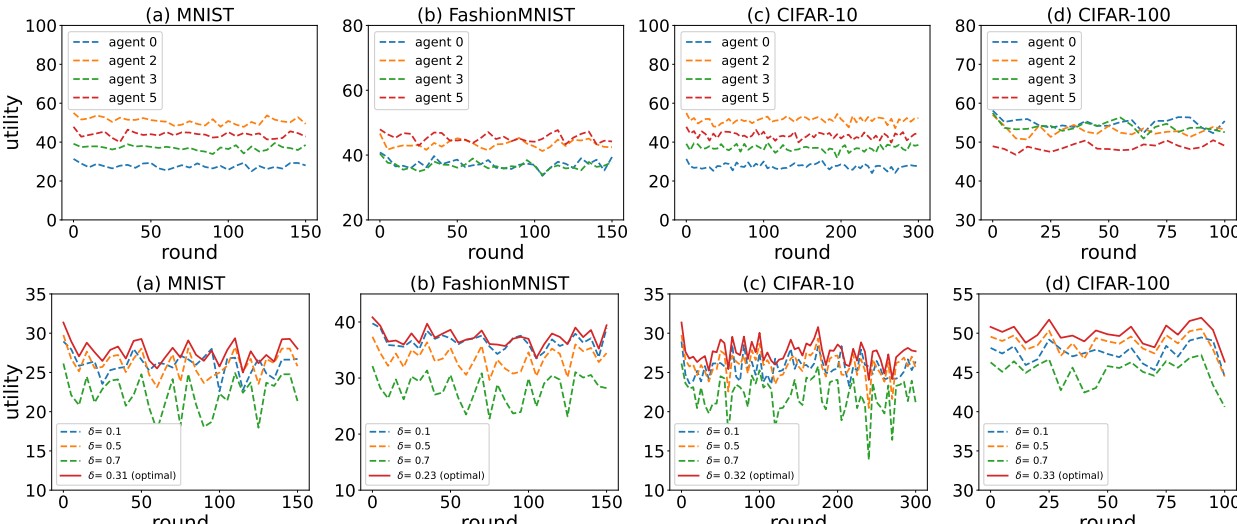

Figure 2: **Upper**: Utility variation across different agents; **Lower**: Utility variation under different non-iid degrees.

**Remark 2.** *(Existence of other possible equilibriums) Note that no one participating is a possible equilibrium in our setting when the marginal cost is too high. In this case, all agents' optimal effort levels will be 0. Another typical free-riding equilibrium does not exist in our setting.*

**Remark 3.** *(The consistency of extending to all agents) Note that inspired by peer prediction, the introduced randomness can serve as an effective tool to circumvent the coordinated strategic behaviors of most agents. However, the consistency of extending to all agents is still satisfied. That is, a consistency exists between randomly selecting one peer agent and the extension that uses the mean of the models.*

Due to space limit, more details about Remark 2 and Remark 3 can be found in Appendix A.10.

**The learner's best response** We will solve the learner's problem shown in Problem 2 in Stage I. Given the optimal effort level $\boldsymbol{e}^*$, the learner only needs to calculate the actual initial payment $Q$ for the payment function. In that case, the payoff can be expressed as Payoff $= g\left(\frac{1}{\text{Bound}(\boldsymbol{e}^*)}\right) - T\sum_{k=1}^{N} f(e_k^*, \boldsymbol{e}_{-k}^*)$. Combined into Problem 2, the optimal solution can be easily computed using a convex program, e.g., the standard solver SLSQP in SciPy (Virtanen et al., 2020).

# 6 Experiments

In this section, we demonstrate existing heterogeneous efforts of non-iid degrees and the performance disparity between agents. We then introduce a logarithmic scoring function to establish an effective incentive mechanism. This mechanism's efficacy is rigorously tested, emphasizing its role in maintaining an equilibrium where no agent can decrease their effort to achieve a higher utility.

**Experimental setup** We adopt the widely utilized federated averaging algorithm (FedAvg) (McMahan et al., 2017) to illustrate the impact of non-iid degrees on effort and incentive issues. We evaluate the performances of several CNN models on four class-balanced image classification datasets: MNIST (LeCun et al., 1998), FashionMNIST (Xiao et al., 2017), CIFAR-10 (Krizhevsky et al., 2009), and CIFAR-100. We evenly partition the whole dataset for agents, and then agents will randomly pick a fixed number of data samples (i.e., 500 or 600) for each round that agents used to train models. Given $N$ agents who fully participate in the training process, the agent will continuously re-sample the data points with a fixed sample size for each round. To simulate the statistical heterogeneity, we carefully construct local datasets for agents by varying the corresponding numbers of distinct classes according to the majority-minority rule. That is, for each agent, a proportion of $\delta$ data samples is from a differed majority class, while the remaining samples are distributed across the rest of the classes following a long-tail distribution (Dai et al., 2023). In particular,

we will employ long-tail distributions to determine the data proportions of minority classes, excluding the majority class. For instance, $\delta = 0.8$ means that 80% of the data samples are from one class (label), and the remaining 20% belong to the other classes (labels). In this way, we can use this metric $\delta$ to measure the non-iid degree, capturing the idea of Wasserstein distance.

## 6.1 Heterogeneous Efforts on Non-iid Degrees

**Heterogeneous efforts** By default, we use a long-tail distribution to determine the data proportions of minority classes, excluding the majority class. Figure 1 demonstrates the performances of three instantize distributions with different $\delta_k$s on four datasets, respectively. In Figure 1, it is apparent that the degree of non-iid has a substantial impact on the actual performance. The greater the heterogeneity in the data, the worse the performance. The default non-iid degree is 0.5. Specifically, $\delta = 0.1$ corresponds to the iid setting for MNIST, FashionMNIST, and CIFAR-10 datasets, where we equally distribute the remaining samples into other classes.

**Performance comparison with peers** We depict the test accuracy of the local models from two agents with different fixed non-iid degrees to show the performance gap between them caused by statistical heterogeneity. In particular, we set the non-iid degree of two agents as 0.1 and 0.9, respectively. Figure 1 illustrates the existing performance gap between two agents. The performance gap can be attributed to the heterogeneous efforts of the data, thereby indicating the feasibility of utilizing it as an effective measure for designing payment functions. Note that many works prefer using the number of classes $p$ within each client's local dataset. as a non-iid metric for data partitioning (Yang et al., 2022). Additional results of this popular setting are provided in Appendix B.2.

## 6.2 Implementing Incentive Mechanism

**Scoring function** We first present a scoring structure as the payment function $f(\cdot)$: a logarithmic function with natural base $e$. For simplicity, the cost functions are designed as classical linear functions. Please refer to Appendix B.3 for more details. We can derive and obtain the optimal effort level for logarithmic functions, which can be generally rewritten into a general form $\delta_k = f(\delta_{k'}^2) = \frac{1}{c} \pm \sqrt{\frac{1}{c^2} - \delta_{k'}^2 - \frac{\Upsilon}{\Phi}}$, where $\delta_{k'}$ is the non-iid degree of a randomly selected peer.

**Parameter analysis** From Theorem 3, payment function mainly depends on two parameters $\Phi$ and $\Upsilon$, which both depend on learning rate $\eta$, Lipschitz constant $L$ and gradient bound $G$. Recall that we set the learning rate $\eta = 0.01$ as a constant, *i.e.*, $\eta_t = \eta, \forall t$. The detailed analysis of Lipschitz constant $L$ and gradient bound $G$ are presented in Appendix B.3. We can obtain that $\Phi \in [300, 30000]$ and $\Upsilon \in [200, 20000]$, respectively. Then, we set the effort-distance function $\delta_k(e_k)$ as an exponential function here: $\delta_k(e_k) = \exp(-e_k)$.

**The existence of equilibrium** From the theoretical aspect (Theorem 5), a Nash equilibrium in our settings means that no agent can improve their utility by unilaterally changing their invested effort levels. Hence, the utility of agents could be viewed as an indicator of the equilibrium. If the utility of agents remains almost constant as the training progresses, an equilibrium is reached. Here, we aim to evaluate the existence of equilibrium through the examination of utility. As shown in Figure 2, the utilities of four randomly selected agents remain stable, which indicates an equilibrium and also confirms Theorem 5. Then, the observed utility fluctuation mainly stems from the randomness in peer selection.

**Impact on utility under different non-iid levels** Figure 2 provides a detailed example illustrating how utility is affected when an individual unilaterally raises or lowers their effort level. More specifically, we present the possible results of a specific single client when he changes his invested effort levels, however, other clients hold their optimal invested effort levels. To mitigate the impact of randomness, we calculate the average values of the utilities within 5 rounds. Even though the randomness of selecting peers brings fluctuation, the utility achieved by the optimal effort levels is much larger than others in expectation.

# 7    Conclusion

In this paper, we quantify the non-iid degree via the Wasserstein distance, and are the first to prove its significance in the convergence bounds of FL. By utilizing this non-iid metric, we explicitly illustrate the potential connections between the learner and agents regarding convergence results and the generalization error gap. The theoretical findings help us fill the research gap where the current incentive methods focus solely on sample size while overlooking the potential incurred data heterogeneity. Therefore, unlike current game-theoretic methods, our proposed mechanism accounts for the impact of data heterogeneity on data contribution and guarantees truthful reporting from agents through a strict Nash equilibrium. Experiments on real-world datasets validate the efficacy of our mechanism.

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

## Appendix

## A  Omitted Proofs

Before presenting the proofs, we first introduce some annotations.

### A.1  Annotation

Here, we rewrite the expected loss over $k$-th agent's dataset $\mathcal{D}_k$ by splitting samples according to their labels. Accordingly, the local objective $F_k(\boldsymbol{w})$ can be reformulated as:

$$F_k(\mathbf{w}) \triangleq \mathbb{E}_{\zeta^k \in \mathcal{D}_k}[\ell(\mathbf{w}, \zeta^k)] = \mathbb{E}_{(\boldsymbol{x}, y) \sim p^{(k)}}\left[\sum_{i=1}^{I} \ell_i(\mathbf{w}, \boldsymbol{x})\right] = \sum_{i=1}^{I} p^{(k)}(y = i)\mathbb{E}_{\boldsymbol{x}|y=i}[\ell(\mathbf{w}, \boldsymbol{x})]. \tag{3}$$

where $\ell(\cdot, \cdot)$ represents the typical loss function, and datapoint $\zeta^k = (\boldsymbol{x}, y)$ uniformly sampled from dataset $\mathcal{D}_k$, $p^{(k)}(y = i)$ refers to denotes the fraction of datapoints in agent $k$'s local dataset (distribution) that are labeled as $y = i$, the subscript $\boldsymbol{x}|y = i$ indicates all datapoints for which the labels are $i$. Suppose that the model requires T rounds (epochs) to achieve convergence.

Then, for each round (epoch) $t$ on agent $k$, its local optimizer performs SGD as the following:

$$\mathbf{w}_t^k = \mathbf{w}_{t-1}^k - \eta_t \sum_{i=1}^{I} p^{(k)}(y = i)\nabla_{\mathbf{w}}\mathbb{E}_{\boldsymbol{x}|y=i}[\ell(\mathbf{w}_{t-1}^k, \boldsymbol{x})]. \tag{4}$$

Let's assume that the learner aggregates local updates from agents after every $E$ step. For instance, the aggregated weight at the $m$-th synchronization is represented by $\overline{\mathbf{w}}_{mE}$. For convenience, we introduce a virtual aggregated sequence $\overline{\mathbf{w}}_t = \sum_{k=1}^{N} p_k \mathbf{w}_t^k$. The discrepancy between the iterates from agents' average $\overline{\mathbf{w}}_t$ over a single iteration can be quantified using the term $||\overline{\mathbf{w}}_t - \mathbf{w}_t^k||$. The following Theorem 2 will show the upper bound of $||\overline{\mathbf{w}}_t - \mathbf{w}_t^k||$ according to $E$ synchronization interval.

### A.2  Proof of Theorem 2

**Theorem 2** (Bounded the divergence of $\{\mathbf{w}_t^k\}$). Let Assumption 4 hold and $G$ be defined therein, given the synchronization interval $E$ (local epochs), the learning rate $\eta_t$. Suppose that $\nabla_{\mathbf{w}}\mathbb{E}_{\boldsymbol{x}|y=i}[\ell_i(\boldsymbol{x}, \mathbf{w})]$ is $L_{\boldsymbol{x}|y=i}$-Lipschitz, it follows that

$$\mathbb{E}[\sum_{k=1}^{N} p_k ||\overline{\mathbf{w}}_t - \mathbf{w}_t^k||^2] \leq 16(E-1)G^2\eta_t^2(1+2\eta_t)^{2(E-1)} \sum_{k=1}^{N} p_k \underbrace{\left(\sum_{i=1}^{I} |p^{(k)}(y=i) - p^{(c)}(y=i)|\right)^2}_{4\delta_k^2}$$

*Proof.* Without loss of generality, suppose that $mE \leq t < (m+1)E$, where $m \in \mathbb{Z}^+$. Given the definitions of $\overline{\mathbf{w}}_t$ and $\mathbf{w}_t^k$, we have

$$||\overline{\mathbf{w}}_t - \mathbf{w}_t^k||$$

$$= ||\sum_{k'=1}^{N} p_{k'} \mathbf{w}_t^{k'} - \mathbf{w}_t^k||$$

$$= ||\sum_{k'=1}^{N} p_{k'}(\mathbf{w}_{t-1}^{k'} - \eta_t \sum_{i=1}^{I} p^{(k')}(y=i)\nabla_{\mathbf{w}}\mathbb{E}_{\boldsymbol{x}|y=i}[\ell(\mathbf{w}_{t-1}^{k'},\boldsymbol{x})]) - \mathbf{w}_{t-1}^k + \eta_t \sum_{i=1}^{I} p^{(k)}(y=i)\nabla_{\mathbf{w}}\mathbb{E}_{\boldsymbol{x}|y=i}[\ell(\mathbf{w}_{t-1}^k,\boldsymbol{x})]||$$

$$\leq ||\sum_{k'=1}^{N} p_{k'} \mathbf{w}_{t-1}^{k'} - \mathbf{w}_{t-1}^k||$$

$$+ \eta_t ||\sum_{k'=1}^{N} p_{k'} \sum_{i=1}^{I} p^{(k')}(y=i)\nabla_{\mathbf{w}}\mathbb{E}_{\boldsymbol{x}|y=i}[\ell(\mathbf{w}_{t-1}^{k'},\boldsymbol{x})] - \sum_{i=1}^{I} p^{(k)}(y=i)\nabla_{\mathbf{w}}\mathbb{E}_{\boldsymbol{x}|y=i}[\ell(\mathbf{w}_{t-1}^k,\boldsymbol{x})]||$$

(from triangle inequality)

$$\leq ||\sum_{k'=1}^{N} p_{k'} \mathbf{w}_{t-1}^{k'} - \mathbf{w}_{t-1}^k||$$

$$+ \eta_t \sum_{k'=1}^{N} p_{k'} ||\sum_{i=1}^{I} p^{(k')}(y=i)\nabla_{\mathbf{w}}\mathbb{E}_{\boldsymbol{x}|y=i}[\ell(\mathbf{w}_{t-1}^{k'},\boldsymbol{x})] - \sum_{i=1}^{I} p^{(k)}(y=i)\nabla_{\mathbf{w}}\mathbb{E}_{\boldsymbol{x}|y=i}[\ell(\mathbf{w}_{t-1}^k,\boldsymbol{x})]||$$

$$\leq \sum_{k'=1}^{N} p_{k'} ||\mathbf{w}_{t-1}^{k'} - \mathbf{w}_{t-1}^k||$$

$$+ \eta_t \sum_{k'=1}^{N} p_{k'} ||\sum_{i=1}^{I} p^{(k')}(y=i) \cdot [\nabla_{\mathbf{w}}\mathbb{E}_{\boldsymbol{x}|y=i}[\ell(\mathbf{w}_{t-1}^{k'},\boldsymbol{x})] - \nabla_{\mathbf{w}}\mathbb{E}_{\boldsymbol{x}|y=i}[\ell(\mathbf{w}_{t-1}^k,\boldsymbol{x})]]||$$

$$+ \eta_t \sum_{k'=1}^{N} p_{k'} ||\sum_{i=1}^{I} [p^{(k')}(y=i) - p^{(k)}(y=i)] \cdot \nabla_{\mathbf{w}}\mathbb{E}_{\boldsymbol{x}|y=i}[\ell(\mathbf{w}_{t-1}^k,\boldsymbol{x})]||$$

$$\leq \sum_{k'=1}^{N} p_{k'}(1 + \eta_t \sum_{i=1}^{I} p^{(k')}(y=i)L_{\boldsymbol{x}|y=i})||\mathbf{w}_{t-1}^{k'} - \mathbf{w}_{t-1}^k||$$

$$+ \eta_t \sum_{k'=1}^{N} p_{k'} g_{max}(\mathbf{w}_{t-1}^k) \sum_{i=1}^{I} |p^{(k')}(y=i) - p^{(k)}(y=i)|.$$

Recall that $p_k$ indicates the weight of the $k$-th agent such that $p_k \geq 0$ and $\sum_{k=1}^{N} p_k = 1$, which follows FedAvg (McMahan et al., 2017). The last inequality holds because we assume that $\nabla_{\mathbf{w}}\mathbb{E}_{\boldsymbol{x}|y=i}[\ell(\mathbf{w}_t^k,\boldsymbol{x})]$ is $L_{\boldsymbol{x}|y=i}$-Lipschitz, *i.e.*, $||\nabla_{\mathbf{w}}\mathbb{E}_{\boldsymbol{x}|y=i}[\ell(\mathbf{w}_t^{k'},\boldsymbol{x})] - \nabla_{\mathbf{w}}\mathbb{E}_{\boldsymbol{x}|y=i}[\ell(\mathbf{w}_t^k,\boldsymbol{x})]|| \leq L_{\boldsymbol{x}|y=i}||\mathbf{w}_t^{k'} - \mathbf{w}_t^k||$, and denote $g_{max}(\mathbf{w}_t^k) = \max_{i=1}^{I} ||\nabla_{\mathbf{w}}\mathbb{E}_{\boldsymbol{x}|y=i}[\ell(\mathbf{w}_t^k,\boldsymbol{x})]||$.

Subsequently, the term $||\mathbf{w}_{t-1}^{k'} - \mathbf{w}_{t-1}^k||$ illustrated in the above inequality can be further deduced as follows.

$$||\mathbf{w}_{t-1}^{k'} - \mathbf{w}_{t-1}^k||$$

$$= ||\mathbf{w}_{t-2}^{k'} - \eta_{t-1}\sum_{i=1}^{I} p^{(k')}(y=i)\nabla_{\mathbf{w}}\mathbb{E}_{\boldsymbol{x}|y=i}[\ell(\mathbf{w}_{t-2}^{k'}, \boldsymbol{x})] - \mathbf{w}_{t-2}^k + \eta_{t-1}\sum_{i=1}^{I} p^{(k)}(y=i)\nabla_{\mathbf{w}}\mathbb{E}_{\boldsymbol{x}|y=i}[\ell(\mathbf{w}_{t-2}^k, \boldsymbol{x})]||$$

$$\leq ||\mathbf{w}_{t-2}^{k'} - \mathbf{w}_{t-2}^k||$$

$$+ \eta_{t-1}||\sum_{i=1}^{I} p^{(k')}(y=i)\nabla_{\mathbf{w}}\mathbb{E}_{\boldsymbol{x}|y=i}[\ell(\mathbf{w}_{t-2}^{k'}, \boldsymbol{x})] - \eta_{t-1}\sum_{i=1}^{I} p^{(k)}(y=i)\nabla_{\mathbf{w}}\mathbb{E}_{\boldsymbol{x}|y=i}[\ell(\mathbf{w}_{t-2}^k, \boldsymbol{x})]||$$

$$\leq ||\mathbf{w}_{t-2}^{k'} - \mathbf{w}_{t-2}^k|| + \eta_{t-1}\sum_{i=1}^{I} p^{(k')}(y=i)||\nabla_{\mathbf{w}}\mathbb{E}_{\boldsymbol{x}|y=i}[\ell(\mathbf{w}_{t-2}^{k'}, \boldsymbol{x})] - \nabla_{\mathbf{w}}\mathbb{E}_{\boldsymbol{x}|y=i}[\ell(\mathbf{w}_{t-2}^k, \boldsymbol{x})]||$$

$$+ \eta_{t-1}g_{max}(\mathbf{w}_{t-2}^k)\sum_{i=1}^{I} |p^{(k')}(y=i) - p^{(k)}(y=i)|$$

$$\leq (1 + \eta_{t-1}\sum_{i=1}^{I} p^{(k')}(y=i)L_{\boldsymbol{x}|y=i})||\mathbf{w}_{t-2}^{k'} - \mathbf{w}_{t-2}^k|| + \eta_{t-1}g_{max}(\mathbf{w}_{t-2}^k)\sum_{i=1}^{I} |p^{(k')}(y=i) - p^{(k)}(y=i)|.$$

Here, we make a mild assumption that $L = L_{\boldsymbol{x}|y=i} = L_{\boldsymbol{x}|y=i'}, \forall i, i'$, implying that the Lipschitz-continuity remains the same regardless of the classes of the samples. We denote $\alpha_t = 1 + \eta_t L$ for simplicity. Plugging $||\mathbf{w}_{t-1}^{k'} - \mathbf{w}_{t-1}^k||$ into the expression of $||\overline{\mathbf{w}}_t - \mathbf{w}_t^k||$, by induction, we have,

$$||\overline{\mathbf{w}}_t - \mathbf{w}_t^k||$$

$$\leq \sum_{k'=1}^{N} p_{k'}\alpha_t||\mathbf{w}_{t-1}^{k'} - \mathbf{w}_{t-1}^k|| + \eta_t\sum_{k'=1}^{N} p_{k'}g_{max}(\mathbf{w}_{t-1}^k)\sum_{i=1}^{I}|p^{(k')}(y=i) - p^{(k)}(y=i)|$$

$$\leq \sum_{k'=1}^{N} p_{k'}\alpha_t\alpha_{t-1}||\mathbf{w}_{t-2}^{k'} - \mathbf{w}_{t-2}^k|| + \eta_t\sum_{k'=1}^{N} p_{k'}g_{max}(\mathbf{w}_{t-1}^k)\sum_{i=1}^{I}|p^{(k')}(y=i) - p^{(k)}(y=i)|$$

$$+ \eta_{t-1}\sum_{k'=1}^{N} p_{k'}\alpha_t g_{max}(\mathbf{w}_{t-2}^k)\sum_{i=1}^{I}|p^{(k')}(y=i) - p^{(k)}(y=i)|$$

$$\leq \sum_{k'=1}^{N} p_{k'}\prod_{t'=t_0}^{t-1}\alpha_{t'+1}||\mathbf{w}_{t_0}^{k'} - \mathbf{w}_{t_0}^k||$$

$$+ \sum_{k'=1}^{N} p_{k'}\sum_{t'=t_0}^{t-1}\eta_{t'+1}\prod_{t''=t'+1}^{t-1}\alpha_{t''+1}g_{max}(\mathbf{w}_{t'}^k)\sum_{i=1}^{I}|p^{(k')}(y=i) - p^{(k)}(y=i)|.$$

Recall that $mE \leq t < (m+1)E$. When $t_0 = t - t' = mE$, for any $k', k \in \mathcal{N}$, $\overline{\mathbf{w}}_{t_0} = \mathbf{w}_{t_0}^{k'} = \mathbf{w}_{t_0}^{k}$. In this case, the first term of the last inequality $\sum_{k'=1}^{N} p_{k'} \prod_{t'=t_0}^{t-1} \alpha_{t'+1} ||\mathbf{w}_{t'}^{k'} - \mathbf{w}_{t'}^{k}||$ equals 0. Then, we have,

$$
\begin{aligned}
&\mathbb{E}[\sum_{k=1}^{N} p_k ||\overline{\mathbf{w}}_t - \mathbf{w}_t^k||^2] \\
&= \sum_{k=1}^{N} p_k \mathbb{E}[||\overline{\mathbf{w}}_t - \mathbf{w}_t^k||^2] \\
&\leq \sum_{k=1}^{N} p_k \mathbb{E}[|| \sum_{k'=1}^{N} p_{k'} \sum_{i=1}^{I} |p^{(k')}(y=i) - p^{(k)}(y=i)| \sum_{t'=t_0}^{t-1} \eta_{t'+1} \prod_{t''=t'+1}^{t-1} \alpha_{t''+1} g_{max}(\mathbf{w}_{t'}^k)||^2] \\
&\leq \sum_{k=1}^{N} p_k \sum_{k'=1}^{N} p_{k'} \left( \sum_{i=1}^{I} |p^{(k')}(y=i) - p^{(k)}(y=i)| \right)^2 \sum_{t'=t_0}^{t-1} \eta_{t'+1}^2 \prod_{t''=t'+1}^{t-1} \alpha_{t''+1}^2 \mathbb{E}[||g_{max}(\mathbf{w}_{t'}^k)||^2] \\
&\leq (E-1)G^2 \eta_{t_0}^2 \alpha_{t_0}^{2(E-1)} \sum_{k=1}^{N} \sum_{k'=1}^{N} p_k p_{k'} \left( \sum_{i=1}^{I} |p^{(k')}(y=i) - p^{(k)}(y=i)| \right)^2 \\
&\leq 4(E-1)G^2 \eta_t^2 (1 + 2\eta_t L)^{2(E-1)} \sum_{k=1}^{N} \sum_{k'=1}^{N} p_k p_{k'} \left( \sum_{i=1}^{I} |p^{(k')}(y=i) - p^{(k)}(y=i)| \right)^2.
\end{aligned}
\tag{5}
$$

where in the last inequality, $\mathbb{E}[||g_{max}(\mathbf{w}_{t'}^k)||^2] \leq G^2$ because of Assumption 4, and we use $\eta_{t_0} \leq 2\eta_{t_0+E} \leq 2\eta_t$ for $t_0 \leq t \leq t_0 + E$. In order to show the degree of non-iid, we rewrite the last term $\sum_{k=1}^{N} \sum_{k'=1}^{N} p_k p_{k'} \left( \sum_{i=1}^{I} |p^{(k')}(y=i) - p^{(k)}(y=i)| \right)^2$ as follows.

$$
\begin{aligned}
&\sum_{k=1}^{N} \sum_{k'=1}^{N} p_k p_{k'} \left( \sum_{i=1}^{I} |p^{(k')}(y=i) - p^{(k)}(y=i)| \right)^2 \\
&\leq 2 \sum_{k=1}^{N} \sum_{k'=1}^{N} p_k p_{k'} \left( \sum_{i=1}^{I} |p^{(k')}(y=i) - p^{(c)}(y=i)| \right)^2 + 2 \sum_{k=1}^{N} \sum_{k'=1}^{N} p_k p_{k'} \left( \sum_{i=1}^{I} |p^{(k)}(y=i) - p^{(c)}(y=i)| \right)^2 \\
&= 2 \sum_{k'=1}^{N} p_{k'} \left( \sum_{i=1}^{I} |p^{(k')}(y=i) - p^{(c)}(y=i)| \right)^2 + 2 \sum_{k=1}^{N} p_k \left( \sum_{i=1}^{I} |p^{(k)}(y=i) - p^{(c)}(y=i)| \right)^2 \\
&= 4 \sum_{k=1}^{N} p_k \left( \sum_{i=1}^{I} |p^{(k)}(y=i) - p^{(c)}(y=i)| \right)^2 \\
&= 16 \sum_{k=1}^{N} p_k \delta_k^2
\end{aligned}
\tag{6}
$$

where in the first inequality, we use the fact that $||x + y||^2 \leq 2||x||^2 + 2||y||^2$, and $p^{(c)}$ represents the actual reference data distribution in the centralized setting.

Therefore, substituting the term in Eq. (6) into Eq. (5), we have,

$$
\mathbb{E}[\sum_{k=1}^{N} p_k ||\overline{\mathbf{w}}_t - \mathbf{w}_t^k||^2] \leq 64(E-1)G^2 \eta_t^2 (1 + 2\eta_t L)^{2(E-1)} \sum_{k=1}^{N} p_k \delta_k^2.
$$

$\square$

### A.3 Proof of Theorem 3

**Theorem 3** Suppose that the expected loss function $F_c(\cdot)$ also follows Assumption 2, then the upper bound of the generalization loss gap between agent $k$ and $k'$ is

$$F_c(\mathbf{w}^k) - F_c(\mathbf{w}^{k'}) \leq \Phi \delta_k^2 + \Phi \delta_{k'}^2 + \Upsilon \tag{7}$$

where $\Phi = 16L^2 G^2 \sum_{t=0}^{E-1} (\eta_t^2 (1 + 2\eta_t^2 L^2))^t$, $\Upsilon = \prod_{t=0}^{E-1} (1 - 2\eta_t L)^t \frac{2G^2 L}{\mu^2} + \frac{LG^2}{2} \sum_{t=0}^{E-1} (1 - 2\eta_t L)^t \eta_t^2$, and $F_c(\mathbf{w}) \triangleq \mathbb{E}_{z \in \mathcal{D}_c}[l(\mathbf{w}, z)]$ denotes the generalization loss induced when the model $\mathbf{w}$ is tested at the dataset $\mathcal{D}_c$.

*Proof.* Due to the L-smoothness, we have

$$F_c(\mathbf{w}^k) - F_c(\mathbf{w}^{k'}) \leq \langle \nabla F_c(\mathbf{w}^{k'}), \mathbf{w}^k - \mathbf{w}^{k'} \rangle + \frac{L}{2} \|\mathbf{w}^k - \mathbf{w}^{k'}\|^2.$$

By Cauchy-Schwarz inequality and AM-GM inequality, we have

$$\langle \nabla F_c(\mathbf{w}^{k'}), \mathbf{w}^k - \mathbf{w}^{k'} \rangle \leq \frac{L}{2} \|\mathbf{w}^k - \mathbf{w}^{k'}\|^2 + \frac{1}{2L} \|\nabla F_c(\mathbf{w}^{k'})\|^2.$$

Then, due to the L-smoothness of $F_c(\cdot)$ (Assumption 2), we can get a variant of Polak-Łojasiewicz inequality, which follows

$$\|\nabla F_c(\mathbf{w}^{k'})\|^2 \leq 2L(F_c(\mathbf{w}^{k'}) - F_c^*) \leq L^2 \|\mathbf{w}^{k'} - \mathbf{w}^*\|^2.$$

Therefore, we have

$$F_c(\mathbf{w}^k) - F_c(\mathbf{w}^{k'}) \leq L \underbrace{\|\mathbf{w}^k - \mathbf{w}^{k'}\|^2}_{\text{Lemma 6}} + \frac{L}{2} \underbrace{\|\mathbf{w}^{k'} - \mathbf{w}^*\|^2}_{\text{Lemma 7}}.$$

Combined with Lemma 6 and Lemma 7, we complete the proof. $\square$

### A.4 Proof of Lemma 6

**Lemma 6.** *Suppose Assumptions 2 to 3 hold, then we have*

$$\|\mathbf{w}^k - \mathbf{w}^{k'}\|^2 \leq 16LG^2 \sum_{t=0}^{E-1} (\eta_t^2 (1 + 2\eta_t^2 L^2))^t \left( \delta_k^2 + \delta_{k'}^2 \right).$$

*Proof.* For any time-step $t + 1$, we have

$$\|\mathbf{w}_{t+1}^k - \mathbf{w}_{t+1}^{k'}\|^2$$

$$= \|\mathbf{w}_t^k - \eta_t \sum_{i=1}^I p^{(k)}(y = i)\nabla_{\mathbf{w}}\mathbb{E}_{\boldsymbol{x}|y=i}[\ell(\mathbf{w}_t^k, \boldsymbol{x})] - \mathbf{w}_t^{k'} + \eta_t \sum_{i=1}^I p^{(k')}(y = i)\nabla_{\mathbf{w}}\mathbb{E}_{\boldsymbol{x}|y=i}[\ell(\mathbf{w}_t^{k'}, \boldsymbol{x})]\|^2$$

$$\leq \|\mathbf{w}_t^k - \mathbf{w}_t^{k'}\|^2 + \eta_t^2 \|\sum_{i=1}^I p^{(k)}(y = i)\nabla_{\mathbf{w}}\mathbb{E}_{\boldsymbol{x}|y=i}[\ell(\mathbf{w}_t^k, \boldsymbol{x})] - \sum_{i=1}^I p^{(k')}(y = i)\nabla_{\mathbf{w}}\mathbb{E}_{\boldsymbol{x}|y=i}[\ell(\mathbf{w}_t^{k'}, \boldsymbol{x})]\|^2$$

$$\leq \|\mathbf{w}_t^k - \mathbf{w}_t^{k'}\|^2 + 2\eta_t^2 \|\sum_{i=1}^I p^{(k')}(y = i)L_{\boldsymbol{x}|y=i}\left[\nabla_{\mathbf{w}}\mathbb{E}_{\boldsymbol{x}|y=i}[\ell(\mathbf{w}_t^k, \boldsymbol{x})] - \nabla_{\mathbf{w}}\mathbb{E}_{\boldsymbol{x}|y=i}[\ell(\mathbf{w}_t^{k'}, \boldsymbol{x})]\right]\|^2$$

$$+ 2\eta_t^2 \|\sum_{i=1}^I \left(p^{(k)}(y = i) - p^{(k')}(y = i)\right)\nabla_{\mathbf{w}}\mathbb{E}_{\boldsymbol{x}|y=i}[\ell(\mathbf{w}_t^{k'}, \boldsymbol{x})]\|^2$$

$$\leq \|\mathbf{w}_t^k - \mathbf{w}_t^{k'}\|^2 + 2\eta_t^2 \left(\sum_{i=1}^I p^{(k)}(y = i)L_{\boldsymbol{x}|y=i}\right)^2 \|\mathbf{w}_t^k - \mathbf{w}_t^{k'}\|^2$$

$$+ 2L\eta_t^2 g_{max}^2(\mathbf{w}_t^{k'})\left(\sum_{i=1}^I |p^{(k)}(y = i) - p^{(k')}(y = i)|\right)^2$$

$$\leq \left(1 + 2\eta_t^2 \left(\sum_{i=1}^I p^{(k)}(y = i)L_{\boldsymbol{x}|y=i}\right)^2\right)\|\mathbf{w}_t^k - \mathbf{w}_t^{k'}\|^2$$

$$+ 2L\eta_t^2 g_{max}^2(\mathbf{w}_t^{k'})\left(\sum_{i=1}^I |p^{(k)}(y = i) - p^{(k')}(y = i)|\right)^2$$

$$\leq (1 + 2\eta_t^2 L^2)\|\mathbf{w}_t^k - \mathbf{w}_t^{k'}\|^2 + 2L\eta_t^2 G^2\left(\sum_{i=1}^I |p^{(k)}(y = i) - p^{(k')}(y = i)|\right)^2.$$

where the third inequality holds because we assume that $\nabla_{\mathbf{w}}\mathbb{E}_{\boldsymbol{x}|y=i}[\ell(\boldsymbol{x}, \mathbf{w})]$ is $L_{\boldsymbol{x}|y=i}$-Lipschitz continuous, *i.e.*, $||\nabla_{\mathbf{w}}\mathbb{E}_{\boldsymbol{x}|y=i}[\ell(\mathbf{w}_t^k, \boldsymbol{x})] - \nabla_{\mathbf{w}}\mathbb{E}_{\boldsymbol{x}|y=i}[\ell(\mathbf{w}_t^{k'}, \boldsymbol{x})]|| \leq L_{\boldsymbol{x}|y=i}||\mathbf{w}_t^k - \mathbf{w}_t^{k'}||$, and denote $g_{max}(\mathbf{w}_t^{k'}) = max_{k=1}^I ||\nabla_{\mathbf{w}}\mathbb{E}_{\boldsymbol{x}|y=i}[\ell(\mathbf{w}_t^{k'}, \boldsymbol{x})]|||$. The last inequality holds because the above-mentioned assumption that $L = L_{\boldsymbol{x}|y=i} = L_{\boldsymbol{x}|y=i'}, \forall i, i'$, *i.e.*, Lipschitz-continuity will not be affected by the samples' classes. Then, $g_{max}(\mathbf{w}_t^{k'}) \leq G$ because of Assumption 4.

When we focus on the aggregation within a single communication round $E$ analogous to the one-shot federated learning setting, we have

$$\|\mathbf{w}_{E-1}^k - \mathbf{w}_{E-1}^{k'}\|^2$$

$$\leq (1 + 2\eta_t^2 L^2)\|\mathbf{w}_{E-2}^k - \mathbf{w}_{E-2}^{k'}\|^2 + 2L\eta_t^2 G^2\left(\sum_{i=1}^I |p^{(k)}(y = i) - p^{(k')}(y = i)|\right)^2$$

$$\leq \prod_{t=0}^{E-1}(1 + 2\eta_t^2 L^2)^t \|\mathbf{w}_0^k - \mathbf{w}_0^{k'}\|^2 + 2LG^2 \sum_{t=0}^{E-1}(\eta_t^2(1 + 2\eta_t^2 L^2))^t\left(\sum_{i=1}^I |p^{(k)}(y = i) - p^{(k')}(y = i)|\right)^2$$

$$\leq 2LG^2 \sum_{t=0}^{E-1}(\eta_t^2(1 + 2\eta_t^2 L^2))^t\left(\sum_{i=1}^I |p^{(k)}(y = i) - p^{(k')}(y = i)|\right)^2.$$

The last inequality holds because for any agent $k, k' \in [N]$, $\mathbf{w}_0 = \mathbf{w}_0^k = \mathbf{w}_0^{k'}$. In order to obtain the non-iid degree $\delta_k$, similarly, we can rewrite the last term $\left(\sum_{i=1}^I |p^{(k)}(y = i) - p^{(k')}(y = i)|\right)^2$ as follows.

$$\left( \sum_{i=1}^{I} |p^{(k)}(y=i) - p^{(k')}(y=i)| \right)^2$$

$$\leq \left( \sum_{i=1}^{I} |p^{(k)}(y=i) - p^{(c)}(y=k)| + \sum_{i=1}^{I} |p^{(k')}(y=i) - p^{(c)}(y=k)| \right)^2$$

$$\leq 2 \left( \sum_{i=1}^{I} |p^{(k')}(y=i) - p^{(c)}(y=i)| \right)^2 + 2 \left( \sum_{i=1}^{I} |p^{(k)}(y=i) - p^{(c)}(y=i)| \right)^2$$

$$= 8(\delta_k^2 + \delta_{k'}^2).$$

where in the first inequality, we use the fact that $||x+y||^2 \leq 2||x||^2 + 2||y||^2$, and $p^{(c)}$ represents the actual reference data distribution in the centralized setting. $\square$

### A.5 Proof of Lemma 7

**Lemma 7.** *Suppose that Assumption 2 and Assumption 4 hold, then we have*

$$\|\mathbf{w}^{k'} - \mathbf{w}^*\|^2 \leq \prod_{t=0}^{E-1} (1 - 2\eta_t L)^t \frac{4G^2}{\mu^2} + G^2 \sum_{t=0}^{E-1} (1 - 2\eta_t L)^t \eta_t^2.$$

*Proof.* Following the classical idea, we have

$$\mathbb{E}\left[ \left\| \mathbf{w}_{t+1}^{k'} - \mathbf{w}^* \right\|^2 \right]$$

$$= \mathbb{E}\left[ \left\| \Pi_{\mathcal{W}} \left( \mathbf{w}_t^{k'} - \eta_t \hat{\mathbf{g}}_t \right) - \mathbf{w}^* \right\|^2 \right]$$

$$\leq \mathbb{E}\left[ \left\| \mathbf{w}_t^{k'} - \eta_t \hat{\mathbf{g}}_t - \mathbf{w}^* \right\|^2 \right]$$

$$= \mathbb{E}\left[ \left\| \mathbf{w}_t^{k'} - \mathbf{w}^* \right\|^2 \right] - 2\eta_t \mathbb{E}\left[ \left\langle \hat{\mathbf{g}}_t, \mathbf{w}_t^{k'} - \mathbf{w}^* \right\rangle \right] + \eta_t^2 \mathbb{E}\left[ \|\hat{\mathbf{g}}_t\|^2 \right]$$

$$= \mathbb{E}\left[ \left\| \mathbf{w}_t^{k'} - \mathbf{w}^* \right\|^2 \right] - 2\eta_t \mathbb{E}\left[ \left\langle \mathbf{g}_t, \mathbf{w}_t^{k'} - \mathbf{w}^* \right\rangle \right] + \eta_t^2 \mathbb{E}\left[ \|\hat{\mathbf{g}}_t\|^2 \right]$$

$$\leq \mathbb{E}\left[ \left\| \mathbf{w}_t^{k'} - \mathbf{w}^* \right\|^2 \right] - 2\eta_t \mathbb{E}\left[ F\left( \mathbf{w}_t^{k'} \right) - F\left( \mathbf{w}^* \right) + \frac{L}{2} \left\| \mathbf{w}_t^{k'} - \mathbf{w}^* \right\|^2 \right] + \eta_t^2 G^2$$

$$\leq \mathbb{E}\left[ \left\| \mathbf{w}_t^{k'} - \mathbf{w}^* \right\|^2 \right] - 2\eta_t \mathbb{E}\left[ \frac{L}{2} \left\| \mathbf{w}_t^{k'} - \mathbf{w}^* \right\|^2 + \frac{L}{2} \left\| \mathbf{w}_t^{k'} - \mathbf{w}^* \right\|^2 \right] + \eta_t^2 G^2$$

$$= (1 - 2\eta_t L) \mathbb{E}\left[ \left\| \mathbf{w}_t^{k'} - \mathbf{w}^* \right\|^2 \right] + \eta_t^2 G^2.$$

where $\Pi_{\mathcal{W}}()$ means the projection, and the second inequality holds due to L-smoothness. Recall that for a one-shot federated learning setting, we have $\mathbf{w}_0^{k'} = \mathbf{w}_0$. Then, for any $mE \leq t < (m+1)E$, unrolling the recursion, we have,

$$\mathbb{E}\left[ \left\| \mathbf{w}_t^{k'} - \mathbf{w}^* \right\|^2 \right] \leq \prod_{t=0}^{E-1} (1 - 2\eta_t L)^t \mathbb{E}\left[ \left\| \mathbf{w}_0^{k'} - \mathbf{w}^* \right\|^2 \right] + G^2 \sum_{t=0}^{E-1} (1 - 2\eta_t L)^t \eta_t^2$$

$$\leq \prod_{t=0}^{E-1} (1 - 2\eta_t L)^t \frac{4G^2}{\mu^2} + G^2 \sum_{t=0}^{E-1} (1 - 2\eta_t L)^t \eta_t^2.$$

The last inequality holds because of Lemma 8.

**Lemma 8.** *(Lemma 2 of Rakhlin (Rakhlin et al., 2011)). If Assumption 1 and Assumption 2 hold, then*

$$\mathbb{E}[||\mathbf{w}_0 - \mathbf{w}^*||^2] \leq \frac{4G^2}{\mu^2}.$$

$\square$

### A.6 More Examples

This subsection provides several additional examples demonstrating the broad applicability of our Theorem 2 to existing convergence results. To ensure better understanding, we adjust these examples' prerequisites and parameter definitions to fit in our setting, and lay out the key convergence findings from these studies (Yang et al., 2021; Karimireddy et al., 2020). For more details, interested readers can refer to the corresponding references.

**Example 2.** *(Lemma 7 in (Karimireddy et al., 2020)). Suppose the functions satisfy Assumptions 1-4 and $\frac{1}{N}\sum_{i=1}^{N}\|\nabla f_i(\mathbf{w})\|^2 \leq G^2 + B^2\|\nabla f(\mathbf{w})\|^2$ (Bounding gradient similarity). For any step-size (local(global) learning rate $\eta_l(\eta_g)$) satisfying $\eta_l \leq \frac{1}{(1+B^2)8LE\eta_g}$ and effective step-size $\tilde{\eta} \triangleq E\eta_g\eta_l$, the updates of FedAvg satisfy*

$$\mathbb{E}\left\|\mathbf{w}^t - \mathbf{w}^\star\right\|^2 \leq \left(1 - \frac{\mu\tilde{\eta}}{2}\right)\mathbb{E}\left\|\mathbf{w}^{t-1} - \mathbf{w}^\star\right\|^2 + \left(1 - \frac{S}{N}\right)4\tilde{\eta}^2 G^2 - \tilde{\eta}\left(\mathbb{E}\left[f\left(\mathbf{w}^{t-1}\right)\right] - f\left(\mathbf{w}^\star\right)\right)$$
$$+ \left(\frac{1}{ES}\right)\tilde{\eta}^2\sigma^2 + 3L\tilde{\eta}\mathcal{E}_t \tag{8}$$

*where $S$ $(N)$ indicates the number of sampled (total) agents, $E$ is the synchronization interval (local steps) and $\mathcal{E}_t$ is the drift caused by the local updates on the clients defined to be*

$$\mathcal{E}_t \triangleq \underbrace{\frac{1}{EN}\sum_{e=1}^{E}\sum_{i=1}^{N}\mathbb{E}_t\left[\left\|\mathbf{w}_{i,k}^t - \mathbf{w}^{t-1}\right\|^2\right]}_{\text{Divergence term}}$$

*Clearly, we can readily replace the original term $\mathcal{E}_t$ with Theorem 2, and subsequently rewrite the convergence result (Theorem 1 of Karimireddy (Karimireddy et al., 2020)) to introduce the degree of non-iid.*

Prior to presenting Example 3, we initially introduce an extra assumption they have made, which is analogous to Assumption 7.

**Assumption 6.** *(Assumption 3 in (Yang et al., 2021)) (Bounded Local and Global Variances) The variance of each local stochastic gradient estimator is bounded by $\mathbb{E}\left[\left\|\nabla f_i\left(\mathbf{w},\xi^i\right) - \nabla f_i(\mathbf{w})\right\|^2\right] \leq \sigma_L^2$, and the global variability of local gradient is bounded by $\mathbb{E}\left[\|\nabla f_i(\mathbf{w}) - \nabla f(\mathbf{w})\|^2\right] \leq \sigma_G^2, \forall i \in [N]$.*

**Example 3.** *(Theorem 1 in (Yang et al., 2021)). Let local and global learning rates $\eta_L$ and $\eta$ satisfy $\eta_L \leq \frac{1}{8LE}$ and $\eta\eta_L \leq \frac{1}{EL}$. Under Assumptions 2, 4, 6, and with full agent participation, the sequence of outputs $\{\mathbf{w}_k\}$ generated by its algorithm satisfies:*

$$\min_{t\in[T]}\mathbb{E}\left[\|\nabla f(\mathbf{w}_t)\|_2^2\right] \leq \frac{f(\mathbf{w}_0) - f(\mathbf{w}^*)}{c\eta\eta_L ET} + \frac{1}{c}\left[\frac{L\eta\eta_L}{2N}\sigma_L^2 + \frac{5E\eta_L^2 L^2}{2}\left(\sigma_L^2 + 6E\sigma_G^2\right)\right] \tag{9}$$

*where $E$ is the synchronization interval (local steps) and $c$ is a constant. Here, we can define the constant $\sigma_G^2 \triangleq 4G^2\delta_k^2$ using Eq.(10) to introduce the non-iid degree $\delta_k^2$. Note that the above result is obtained within a non-convex setting. One can change the right-hand of Eq. (9) into $\mathbb{E}\left[f(\bar{\mathbf{w}}_T) - f(\mathbf{w}^*)\right]$ by using strong convexity condition $\|\nabla f(\mathbf{w})\|_2^2 \geq 2\mu\left(f(\mathbf{w}) - f^*\right)$.*

**Example 4.** *(Theorem 2 of (Yang et al., 2022)) (AFA-CD with General agent Information Arrival Processes). Suppose that the resultant maximum delay under AFL is bounded, i.e., $\tau \triangleq \max_{t \in [T], i \in \mathcal{M}_t} \{\tau_{t,i}\} < \infty$. Suppose that the server-side and agent-side learning rates $\eta$ and $\eta_L$ are chosen as such that the following conditions are satisfied: $6\eta_L^2 \left(2K_{t,i}^2 - 3K_{t,i} + 1\right) L^2 \leq 1, 180\eta_L^2 K_{t,i}^2 L^2 \tau < 1, \forall t, i$ and $2L\eta\eta_L + 6\tau^2 L^2 \eta^2 \eta_L^2 \leq 1$. Under Assumptions $1-3$, the output sequence $\{\mathbf{w}_t\}$ generated by AFA-CD with general agent information arrival processes satisfies:*

$$\frac{1}{T} \sum_{t=0}^{T-1} \mathbb{E} \left\| \nabla f(\mathbf{w}_t) \right\|^2 \leq \frac{4(f_0 - f_*)}{\eta\eta_L T} + 4\left(\alpha_L \sigma_L^2 + \alpha_G \sigma_G^2\right)$$

*where the constants $\alpha_L$ and $\alpha_G$ are defined as:*

$$\alpha_L = \frac{L\eta\eta_L}{m} \frac{1}{T} \sum_{t=0}^{T-1} \frac{1}{K_t} + \frac{3\tau^2 L^2 \eta^2 \eta_L^2}{m} \frac{1}{T} \sum_{t=0}^{T-1} \frac{1}{K_t} + \frac{15\eta_L^2 L^2}{2} \frac{1}{T} \sum_{t=0}^{T-1} \bar{K}_t,$$

$$\alpha_G = \frac{3}{2} + 45L^2 \eta_L^2 \frac{1}{T} \sum_{t=0}^{T-1} \hat{K}_t^2.$$

*Here*

$$\frac{1}{K_t} = \frac{1}{m} \sum_{i \in \mathcal{M}_t} \frac{1}{K_{t,i}}, \bar{K}_t = \frac{1}{m} \sum_{i \in \mathcal{M}_t} K_{t,i}, \hat{K}_t^2 = \frac{1}{m} \sum_{i \in \mathcal{M}_t} K_{t,i}^2$$

*Obviously, we can simply substitute the original term $\sigma_G^2$ using Eq.(10) to obtain our result.*

### A.7 Connections with Assumption of Bounding Global Gradient Variance

Note that there exists a common assumption in (Yang et al., 2021; 2022; Wang et al., 2019; Stich, 2018) about bounding gradient dissimilarity or variance using constants. To illustrate the broad applicability of the proposed non-iid degree metric, we further discuss how to rewrite this type of assumption by leveraging $\delta_k$, that is, re-measuring the gradient discrepancy among agents, *i.e.*, $\|\nabla F_k(\mathbf{w}) - \nabla F(\mathbf{w})\|^2, \forall k \in [N]$. For completeness, we present it in the following Assumption 7.

**Assumption 7.** *(Bounded Global Variance) The global variability of the local gradient of the loss function is bounded by $\mathbb{E}[\|\nabla F_k(\mathbf{w}) - \nabla F(\mathbf{w})\|^2] \leq \sigma^2, \forall k \in [N]$.*

Then, we will rewrite the above assumption and show that this quantity also has a close connection with the Wasserstein distance $\delta_k$ using the following lemma.

**Remark 4.** *(New form of Assumption 7) Let non-iid degree metric $\delta_k$ be defined in Eq. (1). Then, the global variability of the local gradient of the loss function is bounded by*

$$\|\nabla F_k(\mathbf{w}) - \nabla F(\mathbf{w})\|^2 \leq 4G^2 \delta_k^2, \quad \forall k \in [N]. \tag{10}$$

*Proof.* We have,

$$\|\nabla F_k(\mathbf{w}) - \nabla F(\mathbf{w})\| = \left\| \sum_{\zeta \in \mathcal{D}} \nabla_{\mathbf{w}} \ell(\mathbf{w}, \zeta) \left(p^{(k)}(\zeta) - p^{(c)}(\zeta)\right) \right\|$$

$$\leq \sum_{\zeta \in \mathcal{D}} \|\nabla_{\mathbf{w}} \ell(\mathbf{w}, \zeta)\| \left| p^{(k)}(\zeta) - p^{(c)}(\zeta) \right|$$

$$\leq G \sum_{i=1}^{I} \left| p^{(k)}(y = i) - p^{(c)}(y = i) \right| = 2G\delta_k,$$

where the first inequality holds due to Jensen inequality, and the second inequality holds because of the assumption that $\|\nabla_{\mathbf{w}} l(\mathbf{w}, z)\| \leq G_k \leq G$ for any $i$ and $w$, which analogous to Assumption 4. We also discretely split the training data according to their labels $i, \forall i \in [I]$. Here, we make a mild assumption that the aggregated gradients $\nabla F(\mathbf{w})$ of one communication round can be approximately regarded as the gradients resulting from a reference distribution $p^{(c)}$. Therefore, we can use the distribution $p^{(c)}$ as the corresponding distribution of $\nabla F(\mathbf{w})$. $\square$

## A.8 More Discussions about Assumption 5

Here, we first discuss the core requirement of the payment function. It is easily to think that this assumption holds if $\frac{\partial f}{\partial e_k} > 0$ and $\frac{\partial^2 f}{\partial e_k^2} < 0$. The first-order partial derivative of the payment function is

$$\frac{\partial f}{\partial e_k} = f'\left(\frac{Q}{\Phi\delta_k^2 + \Phi\delta_{k'}^2 + \Upsilon}\right) \cdot \frac{-2\Phi Q\delta_k \delta_k'}{\left(\Phi\delta_k^2 + \Phi\delta_{k'}^2 + \Upsilon\right)^2}.$$

For simplicity, we replace $\delta_k(e_k)$ with $\delta_k$. Similarly, the second-order partial derivative of the payment function is,

$$\begin{aligned}
\frac{\partial^2 f}{\partial e_k^2} =& f''\left(\frac{Q}{\Phi\delta_k^2 + \Phi\delta_{k'}^2 + \Upsilon}\right) \cdot \frac{4\Phi^2 Q^2 \delta_k^2 (\delta_k')^2}{\left(\Phi\delta_k^2 + \Phi\delta_{k'}^2 + \Upsilon\right)^4} \\
&+ f'\left(\frac{Q}{\Phi\delta_k^2 + \Phi\delta_{k'}^2 + \Upsilon}\right) \cdot \frac{-2\Phi Q((\delta_k')^2 + \delta_k \delta_k'')(\Phi\delta_k^2 + \Phi\delta_{k'}^2 + \Upsilon) + 4\Phi^2 Q \delta_k^2 (\delta_k')^2}{\left(\Phi\delta_k^2 + \Phi\delta_{k'}^2 + \Upsilon\right)^3} \\
=& f''\left(\frac{Q}{\Phi\delta_k^2 + \Phi\delta_{k'}^2 + \Upsilon}\right) \cdot \frac{4\Phi^2 Q^2 \delta_k^2 (\delta_k')^2}{\left(\Phi\delta_k^2 + \Phi\delta_{k'}^2 + \Upsilon\right)^4} \\
&+ f'\left(\frac{Q}{\Phi\delta_k^2 + \Phi\delta_{k'}^2 + \Upsilon}\right) \cdot \frac{-2Q\Phi[(\Phi\delta_{k'}^2 + \Upsilon - \Phi\delta_k^2)(\delta_k')^2 + (\Phi\delta_{k'}^2 + \Upsilon + \Phi\delta_k^2)\delta_k \delta_k'']}{\left(\Phi\delta_k^2 + \Phi\delta_{k'}^2 + \Upsilon\right)^3}.
\end{aligned}$$

Here, we discuss two payment function settings in terms of linear function and logarithmic function. In particular, note that parameters $\Phi$ and $\Upsilon$ are both greater than 0, that is, $\Phi > 0$ and $\Upsilon > 0$. More discussions about these two parameters are presented in Section 6.

**Linear function.** If the payment function is a linear non-decreasing function, then $f'(\cdot)$ is a constant $M > 0$, and $f''(\cdot) = 0$. Thus, we can simplify the above constraints as follows.

$$\begin{aligned}
\frac{\partial f}{\partial e_k} &= M \cdot \frac{-2\Phi Q\delta_k \delta_k'}{\left(\Phi\delta_k^2 + \Phi\delta_{k'}^2 + \Upsilon\right)^2} > 0, \\
\frac{\partial^2 f}{\partial e_k^2} &= 0 + M \cdot \frac{-2Q\Phi[(\Phi\delta_{k'}^2 + \Upsilon - \Phi\delta_k^2)(\delta_k')^2 + (\Phi\delta_{k'}^2 + \Upsilon + \Phi\delta_k^2)\delta_k \delta_k'']}{\left(\Phi\delta_k^2 + \Phi\delta_{k'}^2 + \Upsilon\right)^3} < 0.
\end{aligned}$$

Note that $\delta_k' \triangleq \frac{d\delta_k(e_k)}{de_k} < 0$ always holds due to the fact that more effort, less data heterogeneous. With this in mind, there is no need to ask for any extra requirement to satisfy $\frac{\partial f}{\partial e_k} > 0$. Therefore, for a linear non-decreasing payment function, if the inequality $(\Phi\delta_{k'}^2 + \Upsilon - \Phi\delta_k^2)(\delta_k')^2 + (\Phi\delta_{k'}^2 + \Upsilon + \Phi\delta_k^2)\delta_k \delta_k'' > 0$ holds, then Assumption 5 holds.

**Logarithmic function.** Now we consider the case with a logarithmic function. The corresponding inequality can be expressed as follows.

$$\begin{aligned}
\frac{\partial f}{\partial e_k} &= \frac{\Phi\delta_k^2 + \Phi\delta_{k'}^2 + \Upsilon}{Q} \cdot \frac{-2\Phi Q\delta_k \delta_k'}{\left(\Phi\delta_k^2 + \Phi\delta_{k'}^2 + \Upsilon\right)^2} = \frac{-2\Phi\delta_k \delta_k'}{\Phi\delta_k^2 + \Phi\delta_{k'}^2 + \Upsilon} > 0, \\
\frac{\partial^2 f}{\partial e_k^2} &= \frac{\left(\Phi\delta_k^2 + \Phi\delta_{k'}^2 + \Upsilon\right)^2}{Q^2} \cdot \frac{4\Phi^2 Q^2 \delta_k^2 (\delta_k')^2}{\left(\Phi\delta_k^2 + \Phi\delta_{k'}^2 + \Upsilon\right)^4} \\
&\quad + \frac{\Phi\delta_k^2 + \Phi\delta_{k'}^2 + \Upsilon}{Q} \cdot \frac{-2Q\Phi[(\Phi\delta_{k'}^2 + \Upsilon - \Phi\delta_k^2)(\delta_k')^2 + (\Phi\delta_{k'}^2 + \Upsilon + \Phi\delta_k^2)\delta_k \delta_k'']}{\left(\Phi\delta_k^2 + \Phi\delta_{k'}^2 + \Upsilon\right)^3} \\
&= \frac{4\Phi^2 \delta_k^2 (\delta_k')^2}{\left(\Phi\delta_k^2 + \Phi\delta_{k'}^2 + \Upsilon\right)^2} + \frac{-2\Phi[(\Phi\delta_{k'}^2 + \Upsilon - \Phi\delta_k^2)(\delta_k')^2 + (\Phi\delta_{k'}^2 + \Upsilon + \Phi\delta_k^2)\delta_k \delta_k'']}{\left(\Phi\delta_k^2 + \Phi\delta_{k'}^2 + \Upsilon\right)^2} \\
&= 2\Phi \cdot \frac{(3\Phi\delta_k^2 - \Phi\delta_{k'}^2 - \Upsilon)(\delta_k')^2 - (\Phi\delta_{k'}^2 + \Upsilon + \Phi\delta_k^2)\delta_k \delta_k''}{\left(\Phi\delta_k^2 + \Phi\delta_{k'}^2 + \Upsilon\right)^2} < 0.
\end{aligned}$$

Similarly, Assumption 5 holds if $(3\Phi\delta_k^2 - \Phi\delta_{k'}^2 - \Upsilon)(\delta_k')^2 - (\Phi\delta_{k'}^2 + \Upsilon + \Phi\delta_k^2)\delta_k\delta_k'' < 0$.

## A.9 Proof of Theorem 4

**Theorem 4.** (Optimal effort level). Consider agent $k$ with its marginal cost and the payment function inversely proportional to the generalization loss gap with any randomly selected peer $k'$. Then, agent $k$'s optimal effort level $e_k^*$ is:

$$e_k^* = \begin{cases} 0, & \text{if} \quad \max_{e_k \in [0,1]} u_k(e_k, e_{k'}) \leq 0; \\ \hat{e}_k & \text{where} \quad \partial f(\hat{e}_k, e_{k'})/\partial\delta_k(\hat{e}_k) + cd'(\delta_k(\hat{e}_k)) = 0, & \text{otherwise.} \end{cases}$$

*Proof.* Given the definition of utility functions, we have:

$$\partial u_k(e_k)/\partial e_k = \partial f(e_k, e_{k'})/\partial e_k + cd'(\delta_k(\hat{e}_k)). \tag{11}$$

It is easy to see that if $u_k(e_k, e_{k'}) \leq 0$, then agent $k$ will have no motivation to invest efforts, *i.e.*, $e_k^* = 0$. In this regard, it can be categorized into two cases according to the value of the cost coefficient $c$.

**Case 1 (high-cost agent):** If the marginal cost $c$ is too high, the utility of agents could still be less than 0, that is, $\max_{e_k \in [0,1]} u_k(e_k, e_{k'}) \leq 0$. In this case, agent $k$ will make no effort, *i.e.*, $e_k^* = 0$.

**Case 2 (low-cost agent):** Otherwise, there exists a set of effort levels $\hat{e}_k$, which can achieve non-negative utility.

If Assumption 5 holds and there exists a set of effort levels $\hat{e}_k$ with non-negative utility, we can easily derive the optimal effort level by using Eq. (11). And the optimal effort level that maximizes the utility can be obtained when $\partial f(\hat{e}_k, e_{k'})/\partial\hat{e}_k + cd'(\delta_k(\hat{e}_k)) = 0$.

Therefore, we finished the proof of Theorem 4. $\qquad\square$

## A.10 Proof of Theorem 5

Before diving into the proof of Theorem 5, we first address the question of whether the utility functions exhibit well-behaved characteristics, and if not, what conditions must be met to ensure such a property.

**Lemma 9.** *If Assumption 5 holds, the utility function $u_k(\cdot)$ is a well-behaved function, which is continuously differentiable and strictly-concave on $e$.*

*Proof.* Here, we certify that $u_k(e_k)$ is a well-behaved function. According to Assumption 5, we can get the maximum and minimum value of $\partial f(e_k, e_{k'})/\partial e_k$ when $e_k = 0$ and $e_k = 1$, respectively. Note that $\partial f(e_k, e_{k'})/\partial e_{k'}$ is the same as $\partial f(e_k, e_{k'})/\partial e_k$. Therefore,

$$\partial u_k(e)/\partial e_k = \partial f(e_k, e_{k'})/\partial e_k + cd'(\delta_k) \geq [\partial f(e_k, e_{k'})/\partial e_k + cd'(\delta_k)]_{e_k=1} \triangleq d_k^2$$

$$\partial u_k(e)/\partial e_{k'} = \partial f(e_k, e_{k'})/\partial e_{k'} \leq [\partial f(e_k, e_{k'})/\partial e_k + cd'(\delta_k)]_{e_k=0} \triangleq d_{k'}^1$$

Note that the derivative of the utility function $u_k(e)$ with respect to $e_k$ has an identical form to the derivative of $u_k(e)$ with respect to $e_{k'}$. Given this, and in light of the discussion presented in Appendix A.8, it becomes clear that the utility function $u_k(\cdot)$ is continuously differentiable with respect to $e$. Integrating Lemma 9 and the discussion about Assumption 5, it becomes clear that the utility function $u_k(\cdot)$ is not only continuously differentiable but also strictly concave with respect to $e$. Thus, we can finish this part of the proof. $\qquad\square$

We emphasize that it is evident that our utility functions are naturally well-behaved, supporting the existence of equilibrium. To achieve this, we resort to Brouwer's fixed-point theorem (Brouwer, 1911), which yields the existence of the required fixed point of the best response function.

**Theorem 5.** (Existence of pure Nash equilibrium). Denote by $\boldsymbol{e}^*$ the optimal effort level, if the utility functions $u_k(\cdot)$s are well-behaved over the set $\prod_{k\in[N]}[0,1]$, then there exists a pure Nash equilibrium in effort level $\boldsymbol{e}^*$ which for any agent $k$ satisfies,

$$u_k(e_k^*, \boldsymbol{e}_{-k}^*) \geq u_k(e_k, \boldsymbol{e}_{-k}^*), \quad \text{for all } e_k \in [0,1], \forall k \in [N].$$

*Proof.* For a set of effort levels $\boldsymbol{e}$, define the best response function $\boldsymbol{R}(\boldsymbol{e}) \triangleq \{R_k(\boldsymbol{e})\}_{\forall k\in[N]}$, where

$$R_k(\boldsymbol{e}) \triangleq \arg\max_{\tilde{e}_k\in[0,1]} \{u_k(\tilde{e}_k, \boldsymbol{e}_{-k}) \triangleq f(\tilde{e}_k, \boldsymbol{e}_{-k}) - c|\delta_k(0) - \delta_k(e_k)| \geq 0\}, \tag{12}$$

where recall that $f(\tilde{e}_k, \boldsymbol{e}_{-k})$ is the payment function inversely proportional to the performance gap between agent $k$ and a randomly selected agent (peer) $k'$ with their invested data distributions $\mathcal{D}_k(e_k)$, $\mathcal{D}'_k(e_{k'})$, respectively. Notice that, by definition, the payment function is merely relevant to a randomly selected agent (peer) $k'$, not for all agents. Here, without loss of generality, we extend the range from a single agent to all agents, that is, rewrite the payment function in the form of $f(e_k, \boldsymbol{e}_{-k})$.

If there exists a fixed point to the mapping function $R(\cdot)$, *i.e.*, there existed $\tilde{\boldsymbol{e}}$ such that $\tilde{\boldsymbol{e}} \in R(\tilde{\boldsymbol{e}})$. Then, we can say that $\tilde{\boldsymbol{e}}$ is the required equilibrium effort level by the definition of the mapping function. Therefore, the only thing we need to do is to prove the existence of the required fixed points. We defer this part of the proof in Lemma 10 by applying Brouwer's fixed-point theorem.

**Lemma 10.** *The best response function $R(\cdot)$ has a fixed point,* i.e., *$\exists \boldsymbol{e}^* \in \prod_{k=1}^N [0,1]$, such that $\boldsymbol{e}^* = R(\boldsymbol{e}^*)$, if the utility functions over agents are well-behaved.*

*Proof.* First, we introduce the well-known Brouwer's fixed point theorem.

**Lemma 11.** *(Brouwer's fixed-point theorem (Brouwer, 1911)) Any continuous function on a compact and convex subset of $R : \prod_{k\in[N]}[0,1] \to \prod_{k\in[N]}[0,1]$ has a fixed point.*

In our case, Lemma 11 requires that the best response function $R$ on a compact and convex subset be continuous. Notice that the finite product of compact, convex, and non-empty sets $\prod_{k\in[N]}[0,1]$ is also compact, convex, and non-empty. Therefore, $R$ is a well-defined map from $\prod_{k\in[N]}[0,1]$ to $\prod_{k\in[N]}[0,1]$, which is a subset of $\mathbb{R}^N$ that is also convex and compact. Therefore, there is only one thing left to prove: $R$ is a continuous function over $\prod_{k\in[N]}[0,1]$.

More formally, we show that for any $\boldsymbol{\tau} \in \mathbb{R}^N$ with $\|\boldsymbol{\tau}\|_1 \leq 1$, $\lim_{\varepsilon\to 0} |R_k(\boldsymbol{e}+\varepsilon\boldsymbol{\tau}) - R_k(\boldsymbol{e})| = 0$. For simplicity, we define $\boldsymbol{e}' = \boldsymbol{e} + \varepsilon\boldsymbol{\tau}$, $x = R_k(\boldsymbol{e})$, and $x' = R_k(\boldsymbol{e}')$. Integrating Lemma 9 and the discussion about Assumption 5, it becomes clear that the utility function $u_k(\cdot)$ is not only continuously differentiable but also strictly concave with respect to $\boldsymbol{e}$. Therefore, there exists a unique solution $x = R_k(\boldsymbol{e})$ to the first-order condition $\partial u_k(e_k, \boldsymbol{e}_{-k})/\partial e_k = 0$ for each $\boldsymbol{e}$. By the Implicit Function theorem, if $\partial^2 u_k(e_k, \boldsymbol{e}_{-k})/\partial e_k^2 \neq 0$, then the solution to the equation $\partial u_k(e_k, \boldsymbol{e}_{-k})/\partial e_k = 0$ is a function of $\boldsymbol{e}$ that is continuous. Therefore, it is obvious that $\lim_{\varepsilon\to 0} |R_k(\boldsymbol{e} + \varepsilon\boldsymbol{\tau}) - R_k(\boldsymbol{e})| = 0$ if $|\boldsymbol{e}' - \boldsymbol{e}| \leq \varepsilon\|\boldsymbol{\tau}\|_1$ and $\|\boldsymbol{\tau}\|_1 \leq 1$. Therefore, $R$ is continuous over the set $\prod_{k\in[N]}[0,1]$. Overall, the remaining proof follows according to Brouwer's fixed point theorem. $\square$

Therefore, as we previously stated, this fixed point $\boldsymbol{e}^*$ of the best response function $R(\cdot)$ also serves as the equilibria of our framework. $\square$

**Remark 5.** *(Existence of other possible equilibriums) Note that no one participating is a possible equilibrium in our setting when the marginal cost is too high. In this case, all agents' optimal effort levels will be 0. Another typical equilibrium in federated learning, known as free-riding, does not exist in our setting.*

*Proof.* Here, we discuss two typical equilibriums in terms of no one participating and free-riding. Note that no one participating is a possible equilibrium in our setting when the marginal cost is too high. In this case, all agents' optimal effort levels will be 0. Then, we further provide a brief discussion of the

free-riding problem. Given Theorem 5, there is always a Nash equilibrium in effort level that no agent can improve their utility by unilaterally changing their contribution. If all players are rational (and such an equilibrium is unique), then such a point is a natural attractor with all the agents gravitating towards such contributions. Therefore, we can use this property to prove that free-riding will not happen in our setting. To avoid free-riding, a reasonable goal for a mechanism designer is to maximize the payoff of the learner when all players are contributing such equilibrium amounts (Karimireddy et al., 2022). Employing a two-stage Stackelberg game is inherently apt for circumventing the free-riding problem in our paper. Unlike the traditional free-riding scenario where all agents are rewarded directly based on the model's accuracy (represented by the convergence bound in our case, which is the underlying cause of free-riding), our game implements a scoring function. This function rewards agents by using a coefficient $Q$ that depends on the convergence bound. Consequently, the newly introduced coefficient $Q$ decouples the direct connection between the model's performance and the agents' rewards. The learner's optimal response is then determined by solving the Problem 2. This ensures the goal of maximizing the learner's payoff and the uniqueness of the equilibrium, thereby preventing free-riding. $\qquad\square$

**Remark 6.** *(The consistency of extending to all agents) Note that the introduced randomness can serve as an effective tool to circumvent the coordinated strategic behaviors of most agents. Then, the consistency of extending to all agents is still satisfied.*

*Proof.* Note that inspired by peer prediction, randomness serves as an effective tool to circumvent the coordinated strategic behaviors of most agents. There is a consistency between randomly selecting one peer agent and the extension that uses the mean of the models. We can verify this consistency by using the mean of models $\overline{\mathbf{w}}$ returned by all agents to substituting $\mathbf{w}$. Here, we present a high-level sketch of our idea, utilizing inequalities similar to those found in the proof of Theorem 3.

$$
\begin{aligned}
F_c(\mathbf{w}^k) - F_c(\overline{\mathbf{w}}) &\leq \langle \nabla F_c(\overline{\mathbf{w}}), \mathbf{w}^k - \overline{\mathbf{w}} \rangle + \frac{L}{2}\|\mathbf{w}^k - \overline{\mathbf{w}}\|^2 \\
&\leq L\|\mathbf{w}^k - \overline{\mathbf{w}}\|^2 + \frac{L}{2}\|\overline{\mathbf{w}} - \mathbf{w}^*\|^2 \\
&\leq \frac{L}{N}\sum_{k'=1}^{N}\left(\|\mathbf{w}^k - \mathbf{w}^{k'}\|^2 + \frac{1}{2}\|\mathbf{w}^{k'} - \mathbf{w}^*\|^2\right).
\end{aligned}
$$

Then, by integrating the insights from Lemma 6 and Lemma 7, we can get a similar result of Theorem 3. Therefore, the consistency holds. $\qquad\square$

# B Experiment Details and Additional Results

## B.1 Parameter settings in evaluation

Here, we provide a detailed overview for these four image classification tasks, including dataset information, parameter settings and model architectures.

**MNIST (LeCun et al., 1998):** a ten-class image classification task. We use a hand-written digits dataset with 60K grayscale images of size $28 \times 28$. It includes 50,000 training images and 10,000 testing images, distributed evenly across 10 classes. We train a CNN model with two $5 \times 5$ convolution layers. The first layer has 20 output channels and the second has 50, with each layer followed by $2 \times 2$ max pooling.

**FashionMNIST (Xiao et al., 2017):** a ten-class image classification task. This dataset consists of 60K images of 10 fashion categories, each $28 \times 28$ in size. It is split into 50,000 training images and 10,000 testing images, with each class having $6,000$ images. We train a CNN model with two $5 \times 5$ convolution layers, where the first layer is equipped with 16 output channels and the second with 32. Following each convolution layer is a $2 \times 2$ max pooling. The batch-size is 256.

**CIFAR-10 (Krizhevsky et al., 2009):** a ten-class image classification task. Contains 60K $32 \times 32$ color images in 10 classes, with each class having $6,000$ images. The dataset is split into $50,000$ training images

and $10,000$ test images. The CNN model has two $5 \times 5$ convolution layers with 6 and 16 output channels, respectively. Each layer follows a $2 \times 2$ max pooling.

**CIFAR-100 (Krizhevsky et al., 2009):** a twenty-class image classification task. This dataset includes 60K $32 \times 32$ color images across 100 fine classes, with 50,000 images for training and 10,000 for testing. Each fine class comprises 500 training images and 100 test images. The 100 classes are divided into 20 mutually exclusive super-classes. In this classification task, we employ 20 coarse labels due to an insufficient amount of data in each fine label. Due to dataset similarity, the model is set up in the same manner as the one employed for CIFAR-10.

In particular, we apply optimizer Adam (Kingma & Ba, 2015) with 0.9 momentum rate to optimize the model, and use BatchNormalization (Ioffe & Szegedy, 2015) for regularization.

### B.2 Additional Performance Results

In this subsection, we evaluate the heterogeneous efforts using an alternate non-iid metric (the number of classes).

Note that many works prefer using the number of classes as a non-iid metric for data partitioning (Yang et al., 2022), that is, using the number of classes $p$ within each client's local dataset, even though this metric may lack flexibility in adjusting the degree of non-iid. Ideally, agents with local datasets containing more classes are expected to achieve better predictive performance. Here, to illustrate the heterogeneous efforts, we also utilize the number of classes (typical) as a heterogeneity metric, despite its rigidity in altering non-iid degrees. The corresponding results are shown in Figure 3 and Figure 4, which align with the previous findings in Section 6.1.

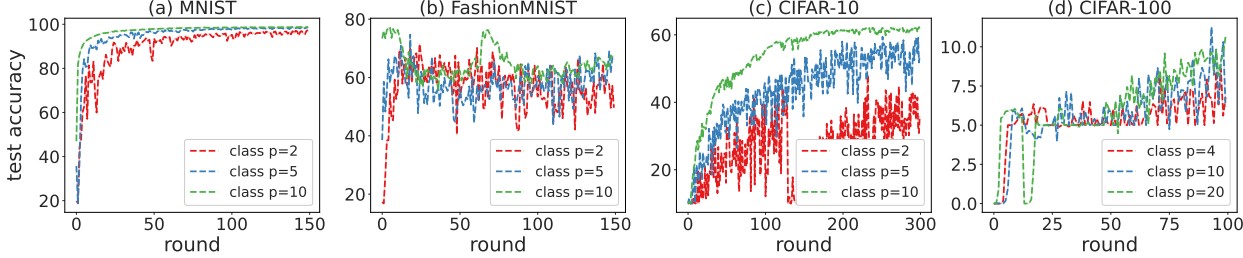

Figure 3: (**The number of classes**) FL training process under different non-iid degrees.

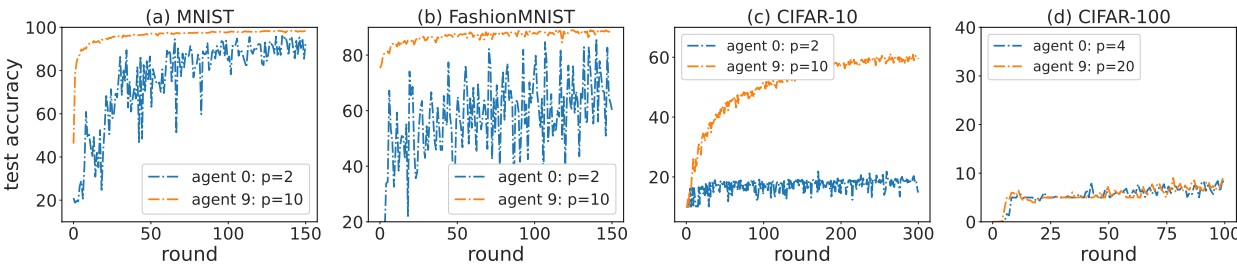

Figure 4: (**The number of classes**) Performance comparison with peers. In default, the non-iid degree is 0.5.

### B.3 Scoring function settings

For linear functions, we have

$$f(e_k, e_{k'}) = \kappa \frac{Q}{\Phi \delta_k^2(e_k) + \Phi \delta_{k'}^2(e_{k'}) + \Upsilon}$$

According to $\partial u_k(e_k)/\partial \delta_k(e_k) = \partial f(e_k, e_{k'})/\partial \delta_k(e_k) + c = 0$, we can derive the optimal effort level by addressing the following equation,

$$\delta_k^4 + 2(\delta_{k'}^2 + \frac{\Upsilon}{\Phi})\delta_k^2 - 2\frac{\kappa Q}{c\Phi}\delta_k + \frac{(\Phi \delta_{k'}^2 + \Upsilon)^2}{\Phi} = 0$$

For brevity, we use $\delta_k$ to represent $\delta_k(e_k)$ as a decision variable. Here, $\delta_{k'}$ can be regarded as a constant. Then, for logarithmic functions, we define it as

$$f(e_k, e_{k'}) = \log(\frac{Q}{\Phi \delta_k^2(e_k) + \Phi \delta_{k'}^2(e_{k'}) + \Upsilon})$$

Similarly, we can easily derive and obtain the optimal effort level by addressing Eq. (13). For simplicity, we take a linear function to simulate $d(\cdot)$ shown in the form of the cost function. Here, we generally rewrite it into a general form $\delta_k \triangleq f(\delta_{k'}^2), \forall k$, to calculate the corresponding $\delta_k$ for agents.

$$\delta_k^2 - \frac{2}{c}\delta_k + \delta_{k'}^2 + \frac{\Upsilon}{\Phi} = 0 \quad \Longrightarrow \quad \delta_k \triangleq f(\delta_{k'}^2) = \frac{1}{c} \pm \sqrt{\frac{1}{c^2} - \delta_{k'}^2 - \frac{\Upsilon}{\Phi}} \tag{13}$$

where $\delta_{k'}$ indicates the non-iid degree of a randomly selected peer. Note that there are two results returned by $f(\delta_{k'}^2)$. Basically, we will choose the smaller one (smaller effort level).

### B.4 Hyperparameter Analysis

Combined with Theorem 3, payment function mainly depends on two parameters $\Phi$ and $\Upsilon$, which both depend on learning rate $\eta$, Lipschitz constant $L$ and the upper bound value of gradient $G$. The detailed analysis of Lipschitz constant $L$ and upper bound value $G$ are presented below.

**Lipschitz constant.** Recall that we focus more on the image classification problem with classical cross-entropy loss. In general, the Lipschitz constant of the cross-entropy loss function is determined by the maximum and minimum probabilities of the output of the network. Specifically, for a neural network with the output probability distribution $\hat{y}$ and true distribution $y$, the Lipschitz constant of the cross-entropy loss function with respect to the L1-norm can be lower-bounded as

$$L \geq \sup\{\|\frac{\partial CE(\hat{y}_n, y_n)}{\partial \hat{y}_n}\|, \forall x_n\} \Longrightarrow L \geq \max_{i \in I}\{|\frac{y_i}{\hat{y}_i}|\}$$

where $\{(x_n, y_n)\}_{n=1}^N$ indicates $N$ samples, and $i \in I$ denotes the index of possible labels. We assume that the corresponding averaged error rate of other labels is 0.01 approximately. Consider the worse case where the ML model misclassifies one sample, which means the probability that the predicted label is the true label is equal to the average error rate. Then, we can derive the bound of the Lipschitz constant using $L \geq \max_{i \in I}\{|\frac{p_i}{q_i}|\} = \frac{1}{0.01} = 100$. Here, we set the default value of $L$ as 100. Recall that the learning rate $\eta = 0.01$, and assume $\eta_t = \eta, \forall t$. Therefore, we can simplify parameter $\Phi$ as

$$\Phi = 2L^2G^2 \sum_{t=0}^{E-1}(\eta_t^2(1 + 2\eta_t^2 L^2))^t \leq 2L^2G^2 \sum_{t=0}^{E-1} 3\eta^2 = 6EG^2.$$

Similarly, for parameter $\Upsilon$, we have

$$\Upsilon = \prod_{t=0}^{E-1}(1 - 2\eta L)^t \frac{2G^2 L}{\mu^2} + \frac{LG^2}{2}\sum_{t=0}^{E-1}(1 - 2\eta L)^t \eta^2 = \prod_{t=0}^{E-1}(-1)^t \frac{2G^2 L}{\mu^2} + \frac{L\eta^2 G^2}{2}\sum_{t=0}^{E-1}(-1)^t$$

Here, if $E$ is even, we have $\Upsilon = \frac{2G^2}{\mu}$; otherwise, $\Upsilon = -\frac{2G^2}{\mu} - \frac{\eta G^2}{2}$. In general, we discuss a specific case that $E$ is even, $i.e.$, $\Upsilon = \frac{2G^2}{\mu}$ to guarantee $\Upsilon$ is non-negative. Here, suppose that the convex constant $\mu = 0.01$.

**Gradient bound.** Recall that $G$ is the upper bound of the gradient on a random sample shown in Assumption 4. In practice, the upper bound value of global gradient variance can be evaluated experimentally. More specifically, this value can be obtained from the norms of gradients observed in agents. Figure 5 presents the norm value of gradients of two randomly selected agents, with these values falling within the interval $[0, 10]$, indicating that $G \in [0, 10]$. Another crucial observation is highlighted in Figure 5, which demonstrates the existence of a gradient gap between agents. This can serve as an alternative metric for developing the payment function for FL platforms.

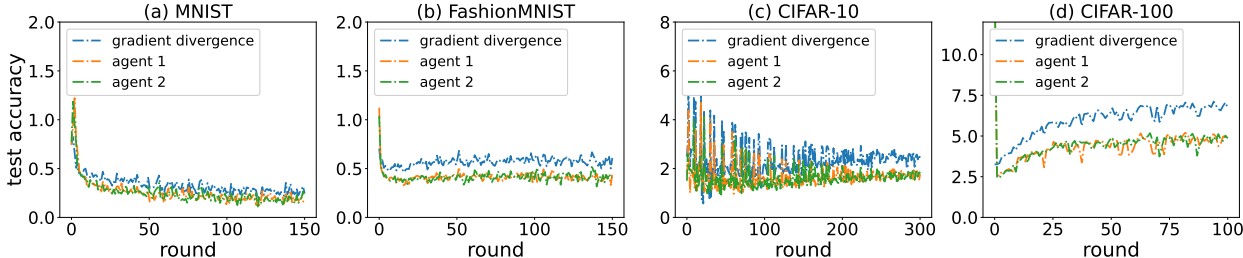

Figure 5: Gradient divergence with peers. The non-iid degree is 0.5.

Based on above parameter analysis, we finally obtain that $\Phi \in [300, 30000]$ and $\Upsilon \in [200, 20000]$, respectively. Due to the Ninety-ninety rule, we set the effort-distance function $\delta_k(e_k)$ as an exponential function here: $\delta_k(e_k) = \exp(-e_k)$.

