# OpenReview forum: "Incentivizing High-quality Participation From Federated Learning Agents"
_TMLR — Rejected by TMLR_

### Review · Reviewer_kVqZ · 2025-09-11

**Summary Of Contributions:**

Currently in federated learning, 1) unmotivated agents might provide no or low-quality data, 2) model doesn’t properly address heterogeneity across diverse data. Thus, this work proposes an incentive framework for agents, considering data heterogeneity to accelerate convergence. For analysis, upper convergence bound using Wasserstein distance is established which is the first such convergence bound for FL, as well as pair-wise generalization error bound using score functions from peer prediction mechanism, filling previous works’ analysis gaps regarding data heterogeneity. Two-stage Stackelberg model examing equilibrium is analyzed. Experiments on real data are performed.

**Additional Comments:**

Besides the convergence bound assuming data heterogeneity, In both theory and especially numerical experiment perspectives, it’s unclear what are the clear and significant contributions of this paper versus existing works. Thus, overall contribution might not be sufficient.

**Audience:**

Yes

**Audience Explanation:**

Might be helpful for federated learning + data heterogeneity.

**Claims And Evidence:**

Yes

**Claims Explanation:**

Yes. Theorems, proofs, experiments, appendices are provided.

**Requested Changes:**

Please add to main paper to clarify as many of your significant contributions as possible, especially any numerical superiority over existing methods. Current contributions are too vague and might not be sufficient. Current numerical superiority is unclear.

---

### Review · Reviewer_LKkP · 2025-09-12

**Summary Of Contributions:**

This paper addresses two significant challenges in federated learning (FL): the lack of incentives for agents to participate and the issue of low-quality data contributions due to heterogeneous effort. The authors propose a novel framework to incentivize high-quality participation and accelerate the convergence of the FL model. The paper's key contributions include:

- Modeling heterogeneous effort: The authors use the Wasserstein distance to quantify the "heterogeneous effort" of different agents. This allows them to reformulate the convergence upper bound of the FL model to account for varying data quality.

- Truthful reporting mechanism: To prevent agents from misrepresenting their data quality, the paper leverages a peer prediction mechanism to create a scoring function that induces truthful reporting.

- Stackelberg game model: This game-theoretic framework provides a formal way to model the interaction between the mechanism designer and the agents, with the goal of optimizing the incentives and the overall model performance.

**Audience:**

Yes

**Audience Explanation:**

The paper tackles a crucial challenge in the real-world deployment of federated learning: how to incentivize participants and ensure high-quality data. The TMLR audience would be interested in findings that provide a robust and mathematically grounded solution to such a problem.

**Claims And Evidence:**

Yes

**Claims Explanation:**

The paper's theoretical claims are supported by detailed mathematical analysis. Experiments on real-world datasets demonstrate the effectiveness of the proposed mechanism in terms of its ability to incentivize improvement.

**Requested Changes:**

-  The Stackelberg game model and the peer prediction mechanism are theoretically sound, but the paper could elaborate on the practical challenges of implementing such a system in a real-world FL setting. For example, how practical is it to calculate the Wasserstein distance in read-world FL setting?

- The notation is sometimes confusing and could be improved. It appears that some subscripts and superscripts are missing. For example:
  - In Example 1, should $w_T$ be $\bar{w}_T$?
  - In Theorem 3 and Lemmas 6 & 7,  should $w^k$ ( and $w^{k'}$) be $w^k_{E-1}$ ( and $w^{k'}_{E-1}$, resp.)?
  - In Example 2, should $w^t$ be $\bar{w}_t$?
  - And so on.

- The analysis focuses on a specific case where $E$ is even and the $\Upsilon$ is non-negative. What happens if this is not satisfied?

---

### Review · Reviewer_T9NE · 2025-09-14

**Summary Of Contributions:**

The authors introduce an incentive-aware framework for Federated Learning (FL) that uses Wasserstein distance to quantify client heterogeneity within convergence bounds, incorporates peer-prediction mechanisms to design scoring functions that encourage truthful reporting, and models server–client interactions as a two-stage Stackelberg game with theoretical guarantees of equilibrium. Experiments on standard image classification datasets are provided to illustrate the approach.

**Strengths**

* Addresses an important and timely problem: incentivizing client participation in FL.
* Provides a clear mathematical formalization linking heterogeneity to convergence via Wasserstein distance.
* Uses game-theoretic reasoning with equilibrium analysis for rigor.
* Creative application of peer-prediction ideas to FL incentive design.

**Weaknesses**

* Novelty is somewhat overstated; Wasserstein-based heterogeneity measures and convergence results have prior precedent.
* Assumptions (e.g., strong convexity, clients reducing heterogeneity by “effort”) are unrealistic for practical FL.
* Experiments rely only on toy datasets with artificial non-iid partitions and lack comparisons to strong baselines such as Shapley-based or auction mechanisms.
* Practical feasibility and scalability of peer-prediction in FL are underexplored.

**Audience:**

Yes

**Audience Explanation:**

Incentive design in Federated Learning is an emerging and underexplored area that directly impacts the deployment of FL in real-world, multi-stakeholder settings. Researchers in machine learning theory, distributed optimization, and mechanism design would find value in the attempt to formalize incentives under data heterogeneity, even if the current formulation and experiments are limited. However, the breadth of interest may be constrained because the work relies heavily on idealized assumptions and toy experiments, which may reduce its appeal to practitioners or empirically driven researchers.

**Claims And Evidence:**

No

**Claims Explanation:**

The submission presents formal theoretical results (e.g., convergence bounds involving the Wasserstein distance, equilibrium existence in a Stackelberg game) and experimental demonstrations on the MNIST, FashionMNIST, and CIFAR datasets. These lend some support to the paper’s main claims that data heterogeneity can be quantified and incorporated into incentive mechanisms, and that the proposed scoring rules lead to equilibrium behavior. However, the evidence is not fully convincing: the theoretical results rely on restrictive assumptions (e.g., strong convexity, agents actively reducing heterogeneity), the experimental setup is limited to toy datasets with artificial partitions, and no comparisons are made against strong existing baselines for incentive mechanisms. Some novelty claims (e.g., “first to prove significance of Wasserstein distance”) are overstated relative to prior work. While the topic is of clear interest to TMLR’s audience, the current evidence base is too narrow and idealized to fully substantiate the bolder claims, and the practical relevance of the proposed mechanism remains unclear.

**Requested Changes:**

- The paper should explicitly compare the use of Wasserstein distance with prior FL work that already employs distributional distances (e.g., total variation, KL, JS). The current “first to prove” claim is overstated and should be moderated with a fairer positioning.

- The experiments need to go beyond toy datasets (MNIST, CIFAR) and artificial non-iid partitions. More realistic FL scenarios (e.g., cross-device text, medical, or speech datasets) and comparisons with strong baselines (Shapley-value–based methods, auction mechanisms, contract-theoretic incentives) are essential to substantiate the claims.

- The notion that agents can “reduce heterogeneity” by effort is underspecified. The paper should clearly define what effort represents in practice (e.g., additional data collection, relabeling, rebalancing) and discuss whether such actions are feasible in real FL systems.

- The design of payment and cost functions currently feels ad hoc. The paper should provide precise definitions, hyperparameter choices, and tuning details, along with code/configurations, to enable reproducibility.

- Discuss whether the results extend to non-convex settings, which are far more common in deep learning. Clarify the implications of relaxing strong convexity and Lipschitz smoothness assumptions.

---

### Decision · Action_Editor_dwrL · 2025-10-22

**Recommendation:** Reject

**Additional Comments:**

The paper proposes an incentive-aware framework for federated learning (FL) that models client heterogeneity via Wasserstein distance and uses peer-prediction–based scoring within a Stackelberg game. While the topic of incentivization in FL is timely and relevant, multiple reviewers raised many key issues: theoretical assumptions are restrictive, limited experiments on small datasets, overstated novelty claims, interpretation of “agent effort”, missing reproducible implementation details, etc. Unfortunately the authors didn't engage with the reviewers to address the concerns. Consequently,  the submission continues to fall short of TMLR’s bar for clarity and validation. We encourage the authors to incorporate the changes and submit again.

**Audience:**

Yes

**Audience Explanation:**

The topic of incentive design in federated learning is relevant to researchers in distributed optimization, privacy-preserving learning, and algorithmic mechanism design.

**Claims And Evidence:**

No

**Claims Explanation:**

The paper provides a formal theoretical treatment as well as empirical results on a few datasets. Howerver, reviewers pointed out issues like theoretical assumptions are restrictive, limited experiments on small datasets, overstated novelty claims, missing reproducible implementation details, etc., which are not addressed.

**Resubmission Of Major Revision:**

The authors may consider submitting a major revision at a later time.